# A Comprehensive Fine-Grained Evaluation of LLMs in Data Race Detection

## Abstract

Data races are a major cause of concurrency-related bugs and have long posed a critical challenge in software engineering. Recent advancements in large language models (LLMs) have inspired researchers to investigate the potential of LLMs in detecting data races. However, the effectiveness of LLMs in this domain still remains largely unexplored, primarily due to the coarse-grained program-level evaluation methodology of existing benchmarks. This article introduces **DRD-Bench**, a novel benchmark, together with **FineEval-Race**, a pioneering evaluation framework, to assess the race detection capabilities of LLMs at the fine-grained individual data race level. DRDBench consists of 1,003 real-world and handcrafted pthreads-based programs, encompassing 549 data races in 226 programs, each annotated with precise line-level race locations. Leveraging this detailed race location information, FineEval-Race establishes fine-grained correspondences between model outputs and ground truth at the level of individual data races, enabling a nuanced evaluation. Based on these fine-grained correspondences, FineEval-Race further evaluates the performance of models under three different response aggregation strategies to investigate the boundary of model capabilities. This methodology not only quantifies LLMs' direct utility in race detection but also provides insights into their genuine understanding of concurrency. We evaluated 25 popular open-source LLMs on DRDBench with FineEval-Race. The evaluation results revealed considerable variation in model performance, with DRDBench presenting a significant challenge to many models. The top-performing reasoning and non-reasoning models, DeepSeek-R1 and DeepSeek-V3, achieved recall of 75.23% and 55.19%, and precision of 75.36% and 54.69%, respectively. These evaluations yield actionable insights. Furthermore, we identify two failure modes shared across models that can cause up to 92% and 98% performance degradation on DeepSeek-R1 and DeepSeek-V3, respectively. We believe that DRDBench and FineEval-Race, coupled with our identified actionable insights and failure modes, will serve as crucial guidance for applying LLMs to race detection and inspire future model training efforts to enhance their comprehension of concurrency.

## 1 Introduction

Writing a bug-free concurrent program is extremely challenging, primarily due to the high non-determinism in thread interleaving (Lu et al., 2008). Data races, defined as two unsynchronized accesses (at least one being a write) to the same shared variable, are a fundamental cause of many concurrency-related bugs. The software engineering community has spent decades investigating heuristic-rule-based and search-based approaches for detecting and verifying data races (Lamport, 1978; Savage et al., 1997; Pavlogiannis, 2020; Cai et al., 2021; Xu et al., 2020; Jeong et al., 2019). However, since the complexity of data race detection and verification is at least NP-complete (Gibbons & Korach, 1997; Mathur et al., 2020), the future of these approaches remains uncertain. A detailed background on this issue is provided in Appendix A.

Recently, neural networks (NNs) and large language models (LLMs) have demonstrated notable proficiency across various tasks, prompting researchers to explore their potential for data race detection. Several benchmarks (Liao et al., 2017; Chen et al., 2023b; TehraniJamsaz et al., 2021) and studies (Chen et al., 2023a; Shen et al., 2025; Alsofyani & Wang, 2024; TehraniJamsaz et al., 2021) have been proposed to evaluate NNs and LLMs in this domain. However, they employ a

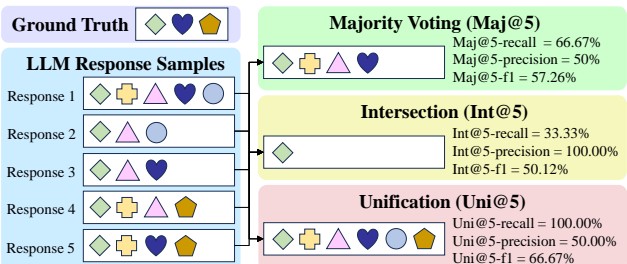

Figure 1: An example of evaluating responses aggregated from $k = 5$ samples via fine-grained majority voting, intersection, and unification strategies at the level of individual data races.

coarse-grained **program-level** evaluation. In their evaluation, the model is instructed to output either (1) a "Yes" or "No" label indicating the presence or absence of data races **in the program**, or (2) a segment of natural language text describing the information of **all data races in the program**. The model's output is then compared to the ground truth through an exact comparison to assess its correctness. This coarse-grained program-level evaluation fails to fully capture the models' capabilities, leaving many important aspects unexamined. For example, when a model fails to generate a correct output, it remains unclear which specific data race contributed to the failure and how far the model's output diverges from the ground truth. Furthermore, existing benchmarks and studies focus on OpenMP-based programs, leaving pthreads-based programs, which are widely used in system-level applications like the Linux kernel, largely unexplored. We provide more background information and a comparison between OpenMP and pthreads programs in Appendix B.

To gain deeper insights about the LLMs' race detection capabilities, this article proposes decomposing the model outputs **at the level of individual data races** and assessing the correctness of **each reported data race** independently. This fine-grained decomposition enables the measurement of **completeness** (the proportion of ground truth data races detected by the LLM), **soundness** (the proportion of correct data races within the LLM's outputs), and the **trade-off** between these two factors. It offers a comprehensive understanding of the LLMs' capabilities in race detection, simultaneously providing insights into their general comprehension of concurrent constructs beyond the race detection task.

Recently, response aggregation techniques, such as self-consistency (Wang et al., 2023; Chen et al., 2024; Wu et al., 2025) and best-of-N (Irvine et al., 2023; Munkhbat et al., 2025; Puri et al., 2025; Parmar et al., 2025), have gained significant attention. They have been proven to be an effective method for improving the performance of LLMs without additional training. Inspired by these achievements, we further investigate the effectiveness of three fine-grained response aggregation strategies in improving the LLMs' race detection capabilities. These aggregation strategies are majority voting, intersection, and unification applied to individual data races reported across multiple LLM responses, which are illustrated in Figure 1. We assess the aggregated model responses for investigating the boundary of model capabilities. This investigation sheds light on the reliability and robustness of LLMs in detecting individual data races, offering deeper insights into their capabilities.

To facilitate the above evaluations, we propose a new benchmark, **DRDBench**, consisting of 1,003 pthreads-based concurrent C programs, among which 226 programs contain 549 precisely annotated data races, and the other 777 programs contain no data races. Based on it, we further introduce a novel evaluation framework, **FineEval-Race**, which (1) rigorously examines the outputs of LLMs to establish correspondences between model outputs and individual ground truth data races for a fine-grained evaluation, and (2) evaluates responses aggregated via three fine-grained strategies to assess the capability boundaries of LLMs in data race detection. We applied DRDBench and FineEval-Race to 25 popular open-source LLMs, including 11 reasoning and 14 non-reasoning models. The evaluation results revealed several key findings. Furthermore, by analyzing common failure cases, we identified two failure modes that can cause significant LLM performance degradation.

In summary, our **contributions** are as follows: (1) We are the first to assess the race detection capabilities of LLMs at the granularity of individual data races. Besides, we are the first to evaluate LLMs in detecting data races on pthreads-based concurrent programs. (2) We introduce **DRDBench**, a new benchmark comprising 1,003 pthreads-based C programs and 549 precisely annotated data races. Complementing this, we present **FineEval-Race**, a novel framework designed for both fine-grained

evaluation of LLM race detection capabilities and investigation of LLM's general comprehension of concurrency. (3) We conduct comprehensive evaluations on 25 popular open-source LLMs, providing the first comprehensive assessment of open-source LLMs in race detection and concurrency comprehension. (4) Our experimental results uncover actionable insights and common failure modes. These findings can offer crucial guidance for the application of LLMs to race detection and inspire future model training efforts aimed at enhancing their understanding of concurrent programs.

## 2 RELATED WORK

**Benchmarks for data race detection.** The SV-Benchmarks (Jain et al., 2025), a famous benchmark that collects various verification tasks used in the annual SV-COMP competition, includes a `NoDataRace` subtrack for evaluating software verifiers' capability to detect data races on pthreads-based programs. Dataracebench (Liao et al., 2017), a commonly used benchmark in data race detection studies (Chen et al., 2023a; Alsofyani & Wang, 2024; TehraniJamsaz et al., 2021; Lin & Liao, 2021; Lin et al., 2018; 2019; Shi et al., 2021), contains 208 OpenMP-based programs for race detection evaluation. Several data race datasets may also be used as benchmarks, including the one constructed by removing synchronization primitives from race-free programs (TehraniJamsaz et al., 2021) and the one obtained by collecting OpenMP-based programs from GitHub (Shen et al., 2025). These benchmarks and datasets only provide basic "Yes" or "No" labels for indicating whether the program contains data races. Such coarse-grained labeling is fundamentally insufficient for evaluating an LLM's genuine comprehension of concurrency because it cannot reliably distinguish whether a model identifies through stochastic guessing or through a true understanding of the intricate semantic details of concurrent program behaviors. As far as we know, the Dataracebench-ML benchmark (Chen et al., 2023a;b), which is an extension of Dataracebench (Liao et al., 2017) by adding race location annotations, is the only benchmark that takes the race location information into evaluation consideration. However, it still treats the data race detection as a program-level binary classification task, as it regards the detection as successful only when the locations of **all races** are correctly predicted. Additionally, its small data size (only 102 data races) and simplicity (OpenMP programs with only 11 to 154 lines of code) may not reflect the real-world scenarios, which can limit the effectiveness of its evaluation results. DRDBench is proposed to address these weaknesses and provide a more comprehensive evaluation.

**Data race detection with neural networks.** With the advancements of neural networks (NNs) and large language models (LLMs) in various software engineering and coding tasks, researchers have begun exploring their potential for data race detection. The researchers have investigated the effectiveness of convolutional neural networks (CNNs) (TehraniJamsaz et al., 2021), prompt engineering and fine-tuning techniques (Chen et al., 2023a), and a parameter-efficient few-shot fine-tuning method (Shen et al., 2025) for classifying race and race-free programs. However, these studies focused on coarse-grained program-level evaluations, i.e., whether a program contains data races or not. They overlooked the evaluation of models in detecting individual data races. We believe FineEval-Race fills this gap and will guide future research in the field.

**LLM reasoning and response aggregation.** Recently, reasoning with LLMs has become a prominent research focus. Reasoning LLMs, such as OpenAI-o1 (OpenAI, 2024b), DeepSeek-R1 (DeepSeek-AI, 2025), and Qwen-3 (Team, 2025a), have shown strong performance across a wide range of tasks. Response aggregation techniques, including self-consistency (Wang et al., 2023; Chen et al., 2024; Wu et al., 2025) and Best-of-N (Irvine et al., 2023; Munkhbat et al., 2025; Puri et al., 2025; Parmar et al., 2025), also demonstrate effectiveness in further improving model performance on reasoning tasks. Since data race detection involves reasoning about code concurrency, we particularly investigate the race detection capability of reasoning models. Besides, we also evaluate the impact of applying response aggregation strategies on LLMs. This is for assessing the boundary of the models' data race reasoning capabilities.

**Evaluating NNs and LLMs for code execution comprehension.** Several recent studies have investigated the capabilities of NNs and LLMs in understanding code executions. Most related works focus on output prediction (Bieber et al., 2020; 2022; Liu et al., 2023; Ding et al., 2024b), while some others examine the intermediate results (Ding et al., 2024a). Benchmarks such as HumanEval (Chen et al., 2021), MBPP (Austin et al., 2021), CRUXEval (Gu et al., 2024), and LiveCodeBench (Jain et al., 2024) have been proposed for this purpose. However, a critical limitation of most existing code

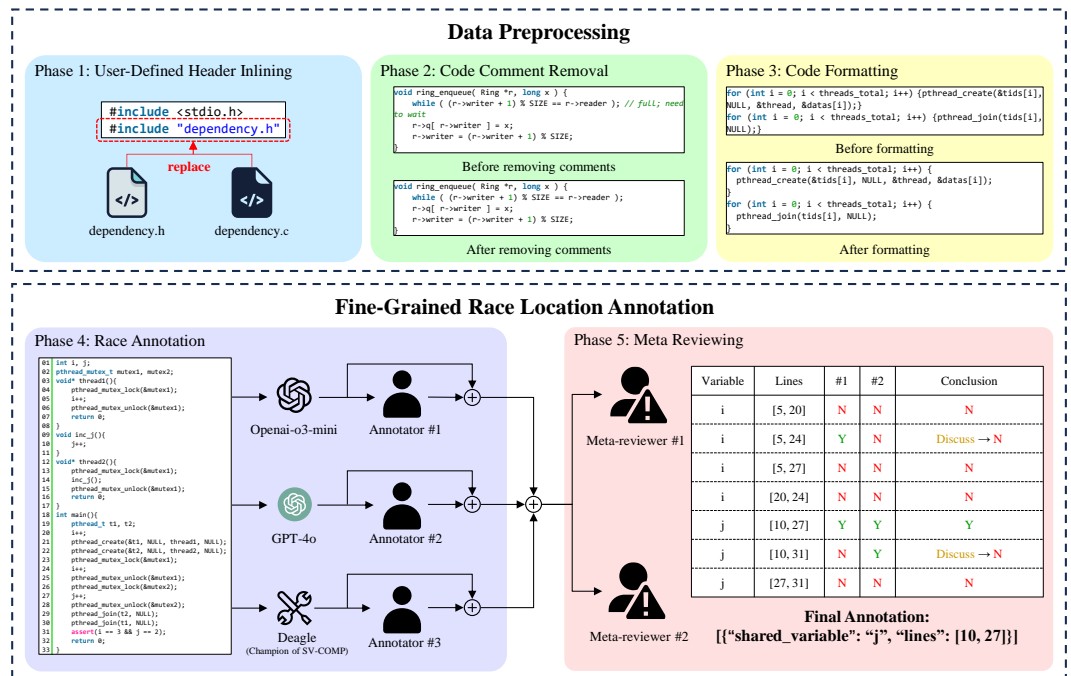

Figure 2: The pipeline of DRDBench construction.

comprehension benchmarks is their sole focus on sequential program execution. Data race detection, in contrast, requires understanding the programming logic of execution under concurrent scenarios. Therefore, DRDBench and FineEval-Race offer a unique and necessary perspective for evaluating code execution comprehension of LLMs within the domain of concurrent programs.

## 3 THE CONSTRUCTION OF DRDBENCH

DRDBench consists of 1,003 pthread-based programs from the `NoDataRace` subtrack of the SV-Benchmarks (Jain et al., 2025), including 226 programs that feature 549 data races and 777 programs that are free of data races. The programs are sourced from Linux drivers and various real-world projects, including Goblint, C-DAC, Deagle, DIVINE, and CProver. The program sizes range from 14 to 624 lines of code, with each program containing between 0 to 30 data races. We provide additional statistics of these programs and illustrate two examples in Appendix C. These programs are originally labeled with a binary flag, indicating whether they are race-free ("Yes") or contain data races ("No"). To construct DRDBench, we manually annotate the precise locations of data races within all programs originally labeled "No". Conversely, the "Yes" (race-free) programs are added directly to the benchmark without further annotation. For each data race, two types of annotations are provided: (1) the name of the shared variable, and (2) the line numbers of the two involved memory accesses. An example annotation is shown at the lower right corner of Figure 2.

The pipeline for the benchmark construction is organized into five distinct phases as illustrated in Figure 2. The three preprocessing phases are introduced to eliminate potential confounding factors that could obscure the assessment of model capabilities in reasoning about concurrent constructs. However, there exists a potential risk that the preprocessing may change the original code patterns and consequently lead to biased evaluations and conclusions. To rigorously assess the influence of such risk, we conduct three ablation studies, detailed in Appendices D, D, and F. These studies demonstrate that although the preprocessing can simplify the race detection task and consequently lead to improved model performance, the impact is limited because performance gains do not exceed 15%. Additionally, we compare model performance on the preprocessed programs against that on the original format programs to further measure the combined influence of all three preprocessing phases. This comparison is presented in Appendix G. This experiment validates that omitting the preprocessing phases does not alter the core insights derived from our main evaluation in Section 5.

In the following, we detail each phase.

(1) **User-defined header inlining**. We first flatten multi-file C programs into the single-file format by replacing `#include` macros of user-defined headers with their corresponding content. The `#include` macros for system headers are left unchanged because LLMs should have learnt system headers during the pre-training. This simplifies the file structure, allowing models to focus on the core concurrent constructs of the source code. Besides, it simplifies the assessment of model outputs by allowing data races to be precisely located by line numbers.

(2) **Code comment removal.** We remove all comments from the source code. This is conducted by replacing the comment lines and blocks with corresponding spaces and blank lines. We ensure that the code structure is unaffected in this phase. This crucial step prevents LLMs from exploiting the information in code comments, thereby ensuring that the evaluation results genuinely reflect the models' own comprehension of the concurrent programming logic.

(3) **Code formatting.** In the final preprocessing phase, we apply formatting operations to the code to ensure uniformity in indentation, spacing, and line break styles across all programs. This ensures that each code line contains at most one statement. It guarantees that all memory accesses occurring on the same line are under identical synchronization scope, which is necessary for line numbers to be a sufficient mechanism for precisely locating data races. This is achieved using the `clang-format` tool configured with the `microsoft` style. Crucially, we manually review the formatted code to verify that the program's functionality and logic remained equivalent to the original version.

(4) **Race annotation.** To facilitate a fine-grained evaluation of LLMs' race detection capabilities, we manually identify and annotate each data race within the programs. This annotation includes two key pieces of information: the **name** of the race-related variable and the **line numbers** of the two involved memory accesses. To reduce the burden on human annotators and improve the quality of the annotations, we leverage three tools: OpenAI-o3-mini (OpenAI, 2025), a leading reasoning LLM, GPT-4o (OpenAI, 2024a), a leading non-reasoning LLM, and Deagle (He et al., 2022), a software verifier, the champion of `NoDataRace` subtrack at the SV-COMP competition for the past four years. Each tool is paired with a human annotator. We first utilize the tools to analyze the program and generate tool annotations. We then allow the human annotators to review the outputs from the tools as hints. This is for the efficiency of race annotation because human annotators usually spend hours on a single program if they annotate from scratch, while the tools' outputs can be useful for helping them gain an initial understanding of the concurrent programs. To mitigate the risk of being misled by tool outputs, we recruit annotators with strong backgrounds (more than 3 years of experience in concurrent programming). Furthermore, we explicitly instruct them not to blindly rely on the tools' annotations. After that, we let each human annotator submit human annotations based on their human understanding of the program semantics. Once all the tool and human annotations are completed, we merge the annotations from the tools and human annotators for further refinement.

(5) **Meta reviewing.** To ensure better quality control, we introduce a meta-reviewing phase aimed at refining the annotations obtained in the previous phase. In this phase, two senior researchers with over five years of experience in data race detection serve as meta-reviewers. Their primary task is to assess the correctness of the previous human and tool annotations. They check whether every annotation is correct and remove the incorrect ones. This verification task can be easier than the annotation task; thus, the meta-reviewers can handle a much greater number of annotations than the annotators in the previous phase. We first instruct each meta-reviewer to evaluate every race annotation and determine its correctness **independently**. If both reviewers agree on a race annotation, it is accepted. If both disagree, it is rejected. If their opinions differ, we let them discuss to reach a consensus. Only annotations that both reviewers agree upon are accepted as the final ground truth. An example of this process is illustrated in Figure 2, where the meta-reviewers initially identify three annotations that could potentially be correct. Their opinions differ on two of them. After a thorough discussion, two annotations are rejected, and only one is accepted. This collaborative review process helps resolve discrepancies and ensures high-quality race annotation.

## 4 FineEval-Race evaluation methodologies

We propose a novel fine-grained evaluation framework, **FineEval-Race**, to evaluate the capabilities of LLMs in data race detection tasks. To begin, we use a carefully crafted zero-shot prompt to

instruct the LLMs to conduct the data race detection and output detailed location information for each identified data race. Due to space limitations, we present the complete prompt in Appendix H. The prompt is structured into 5 sections, including:

**(1) Role and task definition:** This section clearly defines the role and task for the model, specifying that the LLM's goal is to detect data races in the given program.

**(2) Domain-specific knowledge introduction:** This section provides necessary definitions on data races, synchronization primitives, and related concepts to ensure the LLM understands the domain. In Appendix I, we demonstrate that this context is necessary for LLMs to conduct correct race detection; removing this content will lead to significant performance degradation.

**(3) Step-by-step description of the detection procedure:** This section guides the LLM with a step-by-step race detection procedure description, encouraging chain-of-thought reasoning.

**(4) Output format instructions:** This section instructs the model to present its answer in JSON format. Each identified race is represented by three fields, `shared_variable` for the variable name, and `lineA` and `lineB` for the line numbers of the two corresponding memory accesses, respectively. We particularly instruct the model to output an empty JSON object if it feels that the program contains no data race.

**(5) Source code of the program:** The actual code to be analyzed for data race detection, with a line number prepended at the head of each code line to enhance the model's location accuracy.

We parse the output JSON to capture every identified data race. We then examine the data races based on line numbers. The variable name field is excluded from the examination. This exclusion is a trade-off between minimizing false positives and false negatives in ground truth comparison. We discuss the reasons in Appendix J. We retain variable names in annotations and model outputs for clarity and reasoning insights.

A data race report in the model output is considered a **match** with a ground truth data race if their line numbers are identical, regardless of the order. A data race report is **true positive** if it matches a ground truth data race, and it is **false positive** if no such ground truth data race exists. A ground truth data race is **false negative** if no data race report matches it.

Based on the above definitions, we propose several metrics to evaluate the race detection capabilities of LLMs. We utilize the programs that contain data races to conduct a fine-grained evaluation of LLMs' race detection capabilities. We additionally utilize the race-free programs to evaluate the hallucination of LLMs, i.e, whether an LLM reports data races on a race-free program.

On programs that contain data races, we first use the pass rate metric `pass@k` (Kulal et al., 2019) to assess the overall correctness of the model's data race detection. A program is considered **solved** if at least one model output sample achieves (1) all identified data races are *true positive*, and (2) no ground truth data race is *false negative*. The `pass@k` metric calculates the proportion of *solved* programs under the k-sampling setting, as shown below:

$$\texttt{Pass@k} = \frac{\text{The number of programs that contain data races \textit{solved} in } k \text{ output samples}}{\text{The number of programs that contain data races}} \quad (1)$$

For fine-grained evaluation, we further utilize the **recall**, **precision**, and **f1** metrics to evaluate the *completeness*, *soundness*, and *trade-off* of model performance at the level of individual data race:

$$\texttt{Recall} = \frac{\text{The number of \textit{true positive} data races}}{\text{The number of ground truth data races}} \quad (2)$$

$$\texttt{Precision} = \frac{\text{The number of \textit{true positive} data races}}{\text{The number of identified data races (on programs that contain data races)}} \quad (3)$$

$$\texttt{F1} = \frac{2 \times \texttt{Recall} \times \texttt{Precision}}{\texttt{Recall} + \texttt{Precision}} \quad (4)$$

To evaluate the hallucination of LLMs, i.e., detecting a data race on a race-free program, we propose using the **false positive rate** metric (`FPR`), which is defined to be the proportion of race-free programs on which the LLM reports at least one data race. The definition of `FPR` is shown below:

$$\texttt{FPR} = \frac{\text{The number of race-free programs on which the LLM reports at least one data race}}{\text{The number of race-free programs}} \quad (5)$$

Table 1: Evaluation results of 25 popular open-source LLMs on DRDBench (Part 1 of 2).

| Model | S | Pass@1 | Pass@5 | Greedy decoding Recall | Greedy decoding Precision | Greedy decoding F1 | Greedy decoding FPR | Maj@5 Recall | Maj@5 Precision |
|---|---|---|---|---|---|---|---|---|---|
| DeepSeek-R1-671B | #1 47 | #1 **68.14%** | #1 **80.97%** | #1 **75.23%** | #2 75.36% | #1 **75.30%** | #4 13.13% | #1 **77.23%** | #6 79.70% |
| Qwen-QwQ-32B | #1 47 | #2 60.62% | #2 77.43% | #2 65.03% | #1 **77.61%** | #2 70.76% | #2 12.61% | #2 59.02% | #1 **88.28%** |
| Qwen3-Thinking-32B | #3 75 | #5 46.90% | #5 62.39% | #5 48.82% | #3 73.22% | #3 58.58% | #3 12.74% | #4 49.18% | #3 86.82% |
| R1-Distill-Llama-70B | #4 102 | #4 48.23% | #3 73.89% | #6 48.63% | #6 68.46% | #4 56.87% | #9 20.59% | #5 46.27% | #5 82.47% |
| Qwen3-Thinking-30B-A3B | #5 127 | #6 43.81% | #6 61.50% | #7 45.17% | #5 69.47% | #7 54.75% | #7 19.05% | #6 40.98% | #8 78.95% |
| R1-Distill-Qwen2.5-32B | #6 132 | #7 40.71% | #7 59.29% | #8 38.43% | #4 72.01% | #7 50.12% | #10 23.04% | #7 36.07% | #2 86.84% |
| DeepSeek-V3-671B | #7 142 | #3 50.88% | #4 68.14% | #4 55.19% | #9 54.69% | #5 54.94% | #15 49.81% | #3 51.91% | #12 71.43% |
| Qwen2.5-72B | #8 161 | #10 28.32% | #11 40.27% | #10 33.70% | #8 56.23% | #6 42.14% | #6 15.93% | #12 22.59% | #7 78.98% |
| Qwen3-8B | #9 174 | #9 29.65% | #9 45.13% | #9 35.70% | #7 57.48% | #8 44.04% | #14 37.84% | #9 28.96% | #10 75.71% |
| Qwen2.5-Coder-32B | #10 179 | #11 27.88% | #8 47.79% | #13 28.60% | #10 50.81% | #11 36.60% | #11 27.93% | #11 23.68% | #4 84.97% |
| Qwen3-Nothinking-32B | #11 219 | #15 19.03% | #16 26.11% | #14 27.50% | #18 18.11% | #14 21.84% | #5 13.77% | #14 19.49% | #19 48.86% |
| Qwen2.5-32B | #12 228 | #16 17.70% | #13 34.96% | #11 30.05% | #12 40.05% | #12 34.34% | #12 29.73% | #13 22.04% | #11 72.89% |
| Llama-70B | #13 231 | #9 29.20% | #10 42.92% | #3 55.37% | #16 28.73% | #10 37.83% | #19 80.82% | #7 36.07% | #17 52.66% |
| R1-Distill-Llama-8B | #14 259 | #13 20.35% | #12 38.94% | #16 19.85% | #11 42.75% | #13 27.11% | #16 58.69% | #15 15.85% | #13 69.60% |
| Qwen3-Thinking-1.7B | #15 287 | #20 7.96% | #20 19.47% | #20 6.74% | #14 36.27% | #19 11.37% | #13 31.02% | #19 4.55% | #14 65.79% |
| Qwen3-Nothinking-1.7B | #15 287 | #22 3.10% | #21 10.62% | #22 2.73% | #13 38.46% | #22 5.10% | #1 **3.22%** | #21 1.64% | #18 50.00% |
| Qwen3-Nothinking-30B-A3B | #17 297 | #17 16.81% | #18 20.35% | #12 29.51% | #21 12.97% | #17 18.02% | #21 84.43% | #10 26.96% | #22 23.79% |
| Qwen3-Nothinking-8B | #18 301 | #21 3.98% | #23 4.87% | #21 2.91% | #15 34.78% | #21 5.38% | #8 20.08% | #21 1.64% | #21 37.50% |
| R1-Distill-Qwen2.5-7B | #19 313 | #19 8.85% | #18 20.35% | #19 6.92% | #20 17.19% | #20 9.87% | #18 59.85% | #20 3.46% | #9 76.00% |
| Llama-8B | #20 328 | #12 23.89% | #14 27.43% | #15 21.86% | #19 17.67% | #15 19.54% | #25 100.00% | #17 10.38% | #16 55.34% |
| Qwen2.5-Coder-7B | #20 328 | #14 19.47% | #15 26.99% | #18 13.84% | #17 26.21% | #16 18.12% | #22 85.20% | #18 5.83% | #15 60.38% |
| Qwen2.5-7B | #22 333 | #17 16.81% | #17 22.57% | #17 14.39% | #22 11.97% | #18 13.07% | #23 92.41% | #16 12.93% | #20 45.81% |
| R1-Distill-Qwen2.5-1.5B | #23 392 | #24 0.88% | #25 1.33% | #25 0.36% | #24 1.15% | #25 0.55% | #17 59.72% | #25 0.00% | #25 0.00% |
| Qwen2.5-1.5B | #24 414 | #24 0.88% | #24 2.21% | #24 0.73% | #25 0.93% | #24 0.82% | #20 82.63% | #23 0.18% | #24 14.29% |
| Qwen2.5-Coder-1.5B | #25 416 | #23 2.65% | #22 7.08% | #23 1.46% | #23 3.92% | #23 2.12% | #24 97.17% | #23 0.18% | #23 20.00% |
| Deagle | | 50.44% | | 50.27% | 87.34% | 63.82% | 0.26% | | |

To further investigate the capability boundaries of LLMs, we additionally measure the `recall`, `precision`, `f1`, and FPR scores for responses aggregated from $k$ model output samples via three aggregation strategies: **majority voting** (`Maj@k`), **intersection** (`Int@k`), and **unification** (`Uni@k`). These strategies aggregate model outputs at the individual data race level. The majority voting strategy selects data races that appear in at least $\lceil \frac{k}{2} \rceil$ model output samples. The intersection strategy selects those present in all $k$ samples. The unification strategy includes ones that appear in at least one of $k$ samples. An example of these aggregation strategies is shown in Figure 1.

Based on these designs, we propose a synthetic score, denoted as S, for each evaluated model. This score provides an intuitive overall ranking of the models' race detection capabilities. For each model, we calculate the `pass@k` scores at $k = 1$ (using greedy decoding) and $k = 5$. Additionally, we compute the `recall`, `precision`, F1, and FPR scores with greedy decoding, along with `Maj@5`, `Int@5`, and `Uni@5`. This results in a total of 18 unique evaluation scores. We rank all evaluated LLMs according to each evaluation metric independently. The S score for a model $D$ is then calculated as the sum of its rankings across all metrics. Finally, we rank the LLMs in ascending order of the S score, where a lower S score indicates better overall race detection capability.

In the multi-sampling process, we use the corresponding default settings of the hyperparameters temperature, top_k, and top_p as recommended by each model, which are detailed in Appendix K. Our evaluation requires the LLMs to generate a parsable JSON object as the output. If the LLMs do not produce a valid JSON object, we retry with the same prompt. If they still fail to generate a valid JSON object after 10 attempts, we switch to the following settings: temperature = 1.0, top_p = 1.0, and top_k = -1. We then continue the sampling until a parsable JSON object is obtained.

# 5 EXPERIMENT AND ANALYSIS

In experiments, we evaluate the performance of 25 popular open-source LLMs. We choose not to evaluate the closed-source commercial LLMs due to their extremely high financial cost, which is discussed in Appendix L. The evaluated open-source LLMs include DeepSeek-R1 (671B) (DeepSeek-AI, 2025), DeepSeek-V3 (671B) (DeepSeek-AI, 2024), R1's distilled versions on Qwen 2.5 (1.5B, 7B, 32B) (DeepSeek-AI, 2025; Yang et al., 2024), R1's distilled versions on LLama 3.1 (8B, 70B) (DeepSeek-AI, 2025; Grattafiori et al., 2024), Qwen QwQ (32B) (Team, 2025b), Qwen 3 (1.7B, 8B, 32B, 30B-A3B, both thinking and non-thinking modes) (Team, 2025a), Qwen 2.5 (1.5B, 7B, 32B, 72B) (Yang et al., 2024), Qwen 2.5 Coder (1.5B, 7B, 32B) (Hui et al., 2024), and Llama 3.1 (8B, 70B) (Grattafiori et al., 2024). Among these models, DeepSeek-R1, R1's distilled versions on

Table 2: Evaluation results of 25 popular open-source LLMs on DRDBench (Part 2 of 2).

| Model | Maj@5 | | Int@5 | | | | Uni@5 | | | |
|---|---|---|---|---|---|---|---|---|---|---|
| | F1 | FPR | Recall | Precision | F1 | FPR | Recall | Precision | F1 | FPR |
| DeepSeek-R1-671B | [#1] **78.45%** | [#6] 31.66% | [#1] **52.46%** | [#9] 91.14% | [#1] **66.59%** | [#5] 3.17% | [#1] **90.53%** | [#1] **68.27%** | [#1] **77.84%** | [#4] 40.03% |
| Qwen-QwQ-32B | [#2] 70.74% | [#7] 31.79% | [#2] 39.34% | [#5] 94.32% | [#2] 55.53% | [#6] 3.22% | [#2] 79.05% | [#2] 62.90% | [#3] 70.06% | 37.58% |
| Qwen3-Thinking-32B | [#3] 62.79% | [#2] 5.66% | [#5] 26.78% | [#8] 91.30% | [#5] 41.41% | [#2] 0.77% | [#4] 72.86% | [#4] 53.33% | [#6] 61.59% | 47.10% |
| R1-Distill-Llama-70B | [#5] 59.28% | [#10] 37.32% | [#4] 30.24% | [#7] 91.71% | [#13] 45.48% | 5.41% | [#3] 74.32% | [#3] 57.38% | [#7] 64.76% | 47.75% |
| Qwen3-Thinking-30B-A3B | [#6] 53.96% | [#9] 35.14% | [#6] 25.87% | [#10] 91.03% | [#14] 40.28% | 6.05% | [#6] 68.12% | [#5] 51.23% | [#5] 58.48% | 53.54% |
| R1-Distill-Qwen2.5-32B | [#7] 50.97% | [#12] 41.44% | [#8] 19.67% | [#6] 93.91% | [#10] 32.53% | 4.50% | [#8] 59.20% | [#6] 47.45% | [#10] 52.67% | 56.89% |
| DeepSeek-V3-671B | [#4] 60.13% | [#15] 47.23% | [#3] 31.69% | [#11] 90.62% | [#3] 46.96% | [#18] 11.58% | [#3] 77.41% | [#8] 39.61% | [#7] 52.40% | [#15] 73.75% |
| Qwen2.5-72B | [#11] 35.13% | [#5] 29.86% | [#10] 10.93% | [#12] 86.96% | [#10] 19.42% | [#4] 2.16% | [#13] 44.99% | [#9] 34.94% | [#9] 39.33% | [#5] 40.67% |
| Qwen3-Thinking-8B | [#9] 41.90% | [#16] 54.18% | [#14] 9.11% | [#3] 96.15% | [#13] 16.64% | [#9] 4.12% | [#9] 58.83% | [#7] 44.25% | [#8] 50.51% | [#12] 63.19% |
| Qwen2.5-Coder-32B | [#10] 37.04% | [#11] 37.71% | [#12] 10.02% | [#4] 94.83% | [#12] 18.12% | [#10] 4.50% | [#10] 53.19% | [#10] 30.39% | [#11] 38.68% | [#11] 59.85% |
| Qwen3-Nothinking-32B | [#13] 27.86% | [#3] 8.11% | [#11] 10.38% | [#15] 83.82% | [#11] 18.48% | [#2] 0.77% | [#14] 42.08% | [#14] 17.96% | [#13] 25.18% | [#8] 48.39% |
| Qwen2.5-32B | [#12] 33.85% | [#13] 44.27% | [#15] 8.38% | [#13] 86.79% | [#15] 15.28% | [#12] 5.28% | [#11] 48.63% | [#12] 25.28% | [#12] 33.27% | [#13] 63.45% |
| Llama-70B | [#8] 42.81% | [#22] 86.62% | [#9] 14.75% | [#17] 73.64% | [#9] 24.58% | [#17] 10.68% | [#7] 59.56% | [#17] 11.88% | [#18] 19.81% | [#18] 92.28% |
| R1-Distill-Llama-8B | [#14] 25.82% | [#16] 63.84% | [#16] 4.74% | [#14] 86.67% | [#16] 8.98% | [#21] 17.25% | [#12] 45.17% | [#11] 30.02% | [#16] 36.07% | [#16] 82.89% |
| Qwen3-Thinking-1.7B | [#19] 8.52% | [#14] 45.95% | [#20] 0.36% | [#1] **100.00%** | [#20] 0.73% | [#15] 8.11% | [#18] 21.31% | [#13] 20.89% | [#14] 21.10% | [#14] 70.27% |
| Qwen3-Nothinking-1.7B | [#21] 3.17% | [#1] **2.19%** | [#22] 0.00% | [#22] 0.00% | [#22] 0.00% | [#1] **0.00%** | [#21] 6.01% | [#15] 17.46% | [#21] 8.94% | [#1] **8.72%** |
| Qwen3-Nothinking-30B-A3B | [#15] 25.28% | [#17] 57.39% | [#7] 21.31% | [#21] 33.72% | [#8] 26.12% | [#19] 11.84% | [#15] 39.16% | [#19] 8.41% | [#17] 13.85% | 90.60% |
| Qwen3-Nothinking-8B | [#22] 3.14% | [#12] 27.80% | [#17] 1.09% | [#20] 60.00% | [#21] 2.15% | [#8] 3.73% | [#22] 5.65% | [#16] 14.09% | [#22] 8.06% | [#2] 33.72% |
| R1-Distill-Qwen2.5-7B | [#20] 6.62% | [#18] 58.43% | [#21] 0.18% | [#1] **100.00%** | [#21] 0.36% | [#16] 8.37% | [#20] 17.30% | [#17] 12.43% | [#16] 14.47% | [#20] 94.98% |
| Llama-8B | [#17] 17.48% | [#24] 95.11% | [#17] 1.09% | [#16] 75.00% | [#17] 2.15% | [#22] 23.39% | [#16] 30.24% | [#22] 6.75% | [#20] 11.03% | [#24] 100.00% |
| Qwen2.5-Coder-7B | [#18] 10.63% | [#21] 82.50% | [#19] 0.91% | [#18] 71.43% | [#19] 1.80% | [#20] 15.70% | [#17] 21.49% | [#20] 10.36% | [#18] 13.98% | [#23] 98.59% |
| Qwen2.5-7B | [#16] 20.17% | [#23] 90.60% | [#13] 9.29% | [#19] 63.75% | [#14] 16.22% | [#19] 30.50% | [#19] 19.85% | [#17] 10.91% | [#19] 14.08% | [#19] 93.56% |
| R1-Distill-Qwen2.5-1.5B | [#25] 0.00% | [#8] 32.05% | [#22] 0.00% | [#22] 0.00% | [#22] 0.00% | [#7] 3.60% | [#25] 0.73% | [#25] 0.51% | [#25] 0.60% | [#21] 96.27% |
| Qwen2.5-1.5B | [#23] 0.36% | [#20] 81.60% | [#22] 0.00% | [#22] 0.00% | [#22] 0.00% | [#23] 26.00% | [#24] 1.28% | [#24] 0.59% | [#24] 0.81% | [#22] 98.20% |
| Qwen2.5-Coder-1.5B | [#23] 0.36% | [#25] 99.74% | [#22] 0.00% | [#22] 0.00% | [#22] 0.00% | [#25] 72.07% | [#23] 3.83% | [#23] 2.27% | [#23] 2.85% | [#24] 100.00% |

Qwen 2.5 and LLama 3.1, Qwen QwQ, and Qwen 3 (thinking mode) are reasoning models. The others are non-reasoning models. We also report the results of Deagle (He et al., 2022), the champion software verifier of the `NoDataRace` subtrack at the SV-COMP competition for the past four years. Deagle is a rule-based static race detector that translates a program into SMT formulas to conduct the race detection. It targets detecting data races while generating very few false positives. It may still generate a few incorrect results due to the limitations of static analysis in fully capturing the dynamic behaviors, such as dynamic loop exit conditions. Deagle produces deterministic outputs, so we only sample it once. It can serve as a reference for the performance of SOTA non-LLM approaches.

**Main evaluation on open-source LLMs.** The evaluation results are presented in Tables 1 and 2, where the bold and underlined represent the best and second-best performing models evaluated by each metric, respectively. The models are ranked based on their S score, which represents the total of their rankings across all metrics. From these results, we make the following observations:

**(1) Model performance varies significantly, and DRDBench remains a significant challenge for many LLMs**. The two best-performing models, DeepSeek-R1 and Qwen-QwQ, achieved `pass@1` of 68.14% and 60.62%, `f1` (using greedy decoding) of 75.30% and 70.76%, and FPR (using greedy decoding) of 13.13% and 12.61%, respectively. Evaluated by the `f1` scores, they even outperform Deagle, which reason is discussed in Appendix O. Large-scale models, such as Qwen2.5-72B and Llama-70B, performed significantly worse than the two, with `pass@1` of 28.30% and 29.20%, `f1` of 42.14% and 37.83%, and FPR of 15.93% and 80.82%, respectively. Models with 7–8B parameters, which are popular in research, even scored below 25% in `pass@1`, below 30% in `f1`, and most of them have more than 50% in FPR.

**(2) Reasoning models significantly outperform their non-reasoning counterparts.** We observe this phenomenon on DeepSeek, Qwen 3, Qwen2.5, and Llama 3.1 series models, as visualized in Appendix M. For instance, the `pass@1`, `pass@5`, `f1` (greedy decoding), and FPR (greedy decoding) scores of DeepSeek-R1 outperform those of DeepSeek-V3 by 33.92%, 18.83%, 37.06%, and 73.64%, respectively. Appendix N further presents a qualitative analysis comparing the solution chains generated by reasoning and non-reasoning models, illustrating that the observed performance superiority is attributable to improvements in the models' reasoning logic. This highlights the effectiveness of reasoning training in enhancing the model's comprehension of concurrency.

**(3) While response aggregation can improve model performance, the optimal aggregation strategy varies across models.** Compared to greedy decoding, the optimal response aggregation strategy for each model results in an average increase of 19.92% in the F1. However, the best strategy differs among the models: Maj@5 is the optimal for 10 models, including 4 of the top

5 best-performing models; `Uni@5` is the optimal for the other 15 models, helping many weaker models achieve their best performance. We do not consider the `FPR` metrics in this comparison, as `Int@5` should trivially be the optimal strategy when the programs are free of data races. This finding highlights the necessity for further research into developing a new response aggregation strategy to ensure optimal performance across different models. We discuss more about this in Appendix P.

**(4) LLM's race detection is highly unstable but also reveals considerable potential for improvement.** Our evaluation shows that simply increasing the sampling count from 1 to 5 results in a 44.95% improvement in `pass@k` on average. Furthermore, by employing response aggregation strategies, `recall` and `precision` scores can improve by 46.84% (under `Uni@5`) and 78.08% (under `Int@5`) when compared to greedy decoding, respectively. On the one hand, these results demonstrate the instability of LLMs' race detection. On the other hand, they also highlight substantial potential for future improvement. If we can unlock this detection capability, currently observed only in multi-sampling scenarios, within single-sampling contexts, we could substantially enhance LLM performance in data race detection.

**Comparison with coding agents.** Data race detection is conventionally deemed a task suitable for tool-calling agents, primarily because external utilities, such as a debugger, can provide dynamic information to supplement an LLM's static analysis. To investigate the effectiveness of such agents in this domain, we additionally evaluate the OpenHands scaffold (Wang et al., 2025), a popular open-source AI software development platform that equips LLMs with essential tools, including a `bash` terminal, filesystem handlers, and a Python interpreter. Details regarding the evaluation settings and experimental results are presented in Appendix Q due to space limitations. The effectiveness of the agent framework is observed to vary significantly by the underlying backbone model. Specifically, Qwen3-Thinking-32B reduces both its thinking frequency and length when operating within the OpenHands scaffold, inhibiting its performance. Conversely, DeepSeek-V3 leverages task planning and self-reflection in multi-turn conversations to gain improved performance. We provide two examples to illustrate these phenomena in Appendix Q. Critically, **we observe that the LLMs demonstrate no utilization of dynamic code utilities** (such as the `gcc` compiler or `gdb` debugger). This absence of tool utilization may indicate that current LLMs lack the domain-specific knowledge required to associate the analysis of concurrent programs with existing dynamic utilities, resulting in them relying solely on static analysis even when dynamic tools are available.

**Actionable insights for software engineers.** Our evaluation suggests the great potential of LLMs for detecting data races, as the best-performing LLMs can outperform the SOTA static analyzers in certain key metrics. Given the high instability observed in individual LLM predictions, users may aggregate multiple model responses to achieve superior overall detection performance. However, there is no single optimal aggregation strategy. Users must carefully select their aggregation strategy based on their specific engineering requirements (e.g., `Uni@k` for prioritizing recall over precision) and the characteristics of the LLM employed.

**Actionable insights for LLM Researchers.** Both the direct evaluation of LLMs and their performance within the OpenHands agent scaffold consistently demonstrate the critical effectiveness of reasoning capability in the context of race detection. This consistent phenomenon suggests that the reasoning capability is a necessity for fundamentally understanding concurrent programs. Consequently, future LLM training efforts may focus on improving a model's reasoning capabilities on concurrent programming tasks to enhance its overall comprehension of concurrency. Furthermore, the evaluation on the OpenHands scaffold highlights a significant gap: the lack of domain-specific knowledge in current LLMs hinders them from associating the analysis of concurrent programs with dynamic compiling and debugging utilities. Resolving this knowledge gap through specialized pre-training or post-training can be a priority in future model development efforts.

## 6 FAILURE CASE STUDY

To gain deeper insights into the capabilities of LLMs in data race detection, we conduct a manual analysis of common failure cases, including: (1) ground truth data races consistently missed by multiple models, and (2) false positive data races consistently reported by multiple models. For each ground truth data race, we count the number of models that consistently fail to detect it after five samples (`Uni@5`). Similarly, for each false positive data race, we count the number of models that consistently report it across five samples (`Int@5`). The frequency distributions are visualized in Figures 3a and 3b, where the x-axis represents the number of models, and the y-axis represents the

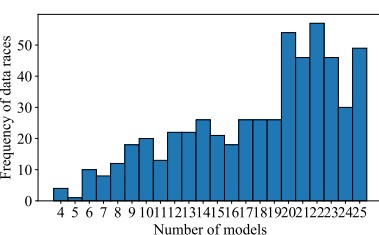

(a) Distribution of false negatives (`Uni@5`).

(b) Distribution of false positives (`Int@5`).

```
01  inline int calculateNext(int s2)
02  {
03      int cnex;
04      do
05          cnex = rand();
06      while (cnex == s2 || cnex == 0);
07      return cnex;
08  }
09  int seed = 1;
10  inline int PseudoRandomUsingAtomic_nextInt()
11  {
12      int read, nexts;
13      assert(seed != 0);
14      atomic_acquire();
15      read = 1; // `read = seed;` leads to failure
16      nexts = calculateNext(read);
17      seed = nexts;
18      atomic_release();
19      return 0;
20  }
21  void *thr1(void *arg)
22  {
23      PseudoRandomUsingAtomic_nextInt();
24      return 0;
25  }
26  int main()
27  {
28      pthread_t t;
29      while (1)
30          pthread_create(&t, 0, thr1, 0);
31  }
```

```
01  void *thr1(void *_) {
02      pthread_mutex_lock(&mutex);      // replace with the following
03      flag1 = 1;                       //    will be fine:
04      while (flag2 == 1)               //
05      {                                // pthread_mutex_lock(&flag1);
06          pthread_mutex_unlock(&mutex); // pthread_mutex_lock(&flag2);
07          pthread_mutex_lock(&mutex);  //
08      }                                //
09      pthread_mutex_unlock(&mutex);    //
10      x = 0;
11      return 0;
12  }
13  void *thr2(void *_) {
14      pthread_mutex_lock(&mutex);      // replace with the following
15      flag2 = 1;                       //    will be fine:
16      while (flag1 == 1)               //
17      {                                // pthread_mutex_lock(&flag2);
18          pthread_mutex_unlock(&mutex); // pthread_mutex_lock(&flag1);
19          pthread_mutex_lock(&mutex);  //
20      }                                //
21      pthread_mutex_unlock(&mutex);    //
22      x = 1;
23      return 0;
24  }
25  int main() {
26      pthread_t t1, t2;
27      pthread_create(&t1, 0, thr1, 0);
28      pthread_create(&t2, 0, thr2, 0);
29      pthread_join(t1, 0);
30      pthread_join(t2, 0);
31      return 0;
32  }
```

(c) Changing a single code line can cause the rate of correctly detecting the data race between lines 13 and 17 to drop from 98% to 32% (DeepSeek-R1) and from 40% to 11% (DeepSeek-V3).

(d) Replacing standard mutex APIs with user-defined synchronization causes the rate of false positive data race reports on variable x to significantly increase from 0% to 92% (DeepSeek-R1) and 2% to 100% (DeepSeek-V3).

Figure 3: Failure case distributions and failure mode illustration.

frequency of either ground truth or false positive data races. For readability, false positive data races that were reported by fewer than five models are omitted from Figure 3b.

These statistics reveal that: (1) even under `Uni@5`, 51.37% (282 out of 549) of ground truth data races are still missed by more than 20 models (Figure 3a), and (2) even under `Int@5`, 22 false positive data races are still consistently reported by more than 10 models. We further investigate these frequently missed or incorrectly reported data races to identify common failure modes.

Among the 282 ground truth data races missed by over 20 models under `Uni@5`, we identified that **multiple occurrences of the same shared variable** is a common cause for these detection failures. Additionally, we observed a common failure mode from the 22 false positive data races consistently reported by over 10 models: the models **fail to understand user-defined synchronization**, even if it has identical semantics as the library APIs. To demonstrate these failure modes, we construct two example programs. The full programs are presented in Appendix R, with brief illustrations in Figures 3c and 3d. We run DeepSeek-R1 and DeepSeek-V3, the best-performing reasoning and non-reasoning models from our evaluation, on these programs 100 times, observing significant performance degradation. We share deeper insights about these failures in Appendix R.

## 7 CONCLUSION

We introduce a new benchmark, DRDBench, and a novel evaluation framework, FineEval-Race, for fine-grained assessment of LLMs' abilities in detecting data races. DRDBench includes 1,003 pthread-based programs with 549 fine-grained data race annotations. FineEval-Race decouples the responses of LLMs to the granularity of individual data races for fine-grained evaluation. We conducted comprehensive experiments on 25 popular open-source LLMs and uncovered several key insights. Additionally, we identified two common failure modes that can lead to significant performance degradation. We believe these findings provide valuable directions for future research.

## 8 ETHICAL DISCUSSION

DRDBench is built upon the open-source benchmark SV-Benchmarks (Jain et al., 2025) and has undergone rigorous ethical reviews and content filtering processes to ensure compliance with the highest ethical standards. We take every precaution to guarantee that the code and data within DRDBench pose no risk of privacy leakage and meet all relevant legal requirements. This commitment ensures that DRDBench is not only an effective benchmark but also one that aligns with ethical guidelines, fostering both technical and ethical integrity.

## 9 REPRODUCIBILITY STATEMENT

In the supplementary material, we provide the datasets, the evaluation scripts, and a README file for illustrating how to reproduce our evaluation results. We have uploaded the supplementary material to the submission site (OpenReview). The README file can also be reviewed online at `https://anonymous.4open.science/r/DRDBench-DE0E`. We believe it can guarantee the reproducibility of our experiments.

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

## A    BACKGROUND ON DATA RACE DETECTION

Data race detection is a prominent research area in the software engineering community. The existing approaches can be primarily divided into two categories: **heuristic-rule-based** approaches and **search-based** approaches.

The two most commonly used race detection techniques within the *heuristic-rule-based* approaches are the **happens-before relation** (Lamport, 1978) and the **lockset discipline** (Savage et al., 1997).

**(1) Happens-before relation.** Most approaches (Flanagan & Freund, 2009; Bond et al., 2010; Elmas et al., 2007; Pozniansky & Schuster, 2007; Serebryany & Iskhodzhanov, 2009) are developed on top of the *happens-before relation*. The *happens-before relation* technique models the chronological orders between critical synchronization operations observed during the program executions as the *happens-before orders*. It then heuristically assumes that those synchronization operations performed chronologically earlier should causally *happen before* later ones. If two operations are not ordered by the *happens-before orders*, they are considered potentially concurrent, and thus a data race may occur between them.

**(2) Lockset discipline.** Some other approaches (Yu et al., 2005; von Praun & Gross, 2001; Choi et al., 2002; Nishiyama, 2004) rely on the *lockset discipline*. They detect data races by checking whether the same mutex lock protects two memory accesses. If different mutexes protect two accesses, they assume the accesses can run concurrently, thereby potentially resulting in a data race.

The widely used Google ThreadSanitizer (Google, 2023) further hybridizes both techniques to achieve effective race detection.

However, these approaches are *unsound*, meaning they may report false data races. This stems from two major limitations: (1) the *happens-before relation* does not fully capture the causal relationships between synchronization operations, and (2) memory accesses may be synchronized through mechanisms other than mutex locks. These limitations often result in false positives in practical scenarios.

Recently, several *search-based* approaches (Mathur et al., 2018; Smaragdakis et al., 2012; Kini et al., 2017; Roemer et al., 2018; Mathur et al., 2021; Pavlogiannis, 2020; Cai et al., 2021; Xu et al., 2020; Jeong et al., 2019) have been proposed, which achieve *soundness*, meaning they avoid reporting false data races. These approaches either (1) carefully model the causal order of memory operations and search for feasible reordering that reveals data races without violating the causal order, or (2) execute the program multiple times under different settings to search for evidence that may expose data races.

However, these approaches suffer from high time complexity. For example, the two most advanced *search-based* approaches, M2 (Pavlogiannis, 2020) and SeqCheck (Cai et al., 2021), have a time complexity of $O(n^4 \log n)$ where $n$ is the number of operations to be analyzed, which limits their scalability and efficiency.

Given the NP-complete nature of data race detection and verification (Gibbons & Korach, 1997; Mathur et al., 2020), the future of heuristic-rule-based and search-based approaches remains uncertain. Motivated by recent advancements in neural networks (NNs) and large language models (LLMs), researchers are increasingly exploring their potential for more effective and efficient data race detection.

## B    BACKGROUND ON OPENMP AND PTHREADS

Parallel programming techniques are critical for improving the performance of applications by leveraging multiple processors or cores. Two widely used frameworks for parallelism are **OpenMP (Open Multi-Processing)**[1] and **pthreads (POSIX threads)**[2]. While both enable concurrent execution, they offer distinct approaches to parallelism. OpenMP provides a high-level, abstraction-based model that simplifies parallelism for shared-memory systems, while pthreads gives developers low-level control over thread management in environments requiring more granular control. Both approaches are widely used, but the choice between them depends on the specific requirements of the application,

---

[1] https://www.openmp.org/
[2] https://man7.org/linux/man-pages/man7/pthreads.7.html

Table 3: Comparison between OpenMP and pthreads.

| Framework | Parallel granularity | Parallelism management | Programming & Complexity | Common use case |
|---|---|---|---|---|
| OpenMP | Loop-level | Automatical | By directives, simple | Scientific computing |
| Pthreads | Thread-level | Manual | By API calls, complex | System-level programming |

```c
01  #include <stdio.h>
02  #include <omp.h>
03
04  #define N 1000
05
06  int main()
07  {
08      int arr[N];
09      int sum = 0;
10
11      for (int i = 0; i < N; i++)
12      {
13          arr[i] = 1;
14      }
15
16      omp_set_num_threads(4);
17
18  #pragma omp parallel for reduction(+ : sum)
19      for (int i = 0; i < N; i++)
20      {
21          sum += arr[i];
22      }
23
24      printf("Sum of array elements: %d\n", sum);
25
26      return 0;
27  }
```

Figure 4: A concurrent program that sums all the elements in an array using 4 threads under the OpenMP framework. The programmer only needs to use the #pragma directive at line 18 for parallelization. OpenMP handles task dispatching and data synchronization automatically.

with OpenMP being preferred for ease of use and pthreads being essential for scenarios where detailed thread control is critical. Figures 4 and 5 present two examples of the same concurrent program written using the OpenMP and pthreads frameworks. Tables 3 summarizes the key differences between OpenMP and pthreads. In the following two subsections, we introduce the technical features of these two frameworks.

## B.1 OPENMP

OpenMP is a widely adopted parallel programming framework that provides an easy-to-use interface for parallelizing applications, primarily targeting shared-memory architecture. It employs compiler directives (#pragma in C/C++) to mark sections of code that should be executed in parallel. OpenMP abstracts the complexity of thread management, allowing developers to focus on the logic of parallelism rather than low-level thread creation, synchronization, and communication.

**Programming model**. OpenMP follows a shared-memory model where multiple threads can access the same shared memory, simplifying data sharing between threads. The primary method for parallelization is through **loop-level parallelism**, where iterations of a loop can be executed concurrently.

**Automated parallelism management**. Although developers can manually configure certain aspects of the parallelism such as the number of threads and data sharing strategies, OpenMP manages most aspects including thread creation, scheduling, and synchronization **automatically**.

**Easy to use**. Developers mainly rely on OpenMP **directives**, e.g., #pragma omp parallel for, to parallelize loops with minimal code changes, making it an accessible tool for parallel programming.

**Use cases**. OpenMP is commonly used in scientific computing, numerical simulations, and data-intensive applications where fine-grained parallelism is needed, and the overhead of managing threads is minimized by the abstraction it provides.

```
01  #include <stdio.h>
02  #include <pthread.h>
03
04  #define N 1000
05  #define NUM_THREADS 4
06
07  int arr[N];
08  int sum = 0;
09  pthread_mutex_t sum_mutex;
10
11  typedef struct
12  {
13      int start_index;
14      int end_index;
15  } ThreadData;
16
17  void *compute_sum(void *arg)
18  {
19      ThreadData *data = (ThreadData *)arg;
20      int local_sum = 0;
21      for (int i = data->start_index; i < data->end_index; i++)
22      {
23          local_sum += arr[i];
24      }
25      pthread_mutex_lock(&sum_mutex);
26      sum += local_sum;
27      pthread_mutex_unlock(&sum_mutex);
28      return NULL;
29  }
30
31  int main()
32  {
33      pthread_t threads[NUM_THREADS];
34      ThreadData thread_data[NUM_THREADS];
35      int segment_size = N / NUM_THREADS;
36      for (int i = 0; i < N; i++)
37      {
38          arr[i] = 1;
39      }
40      pthread_mutex_init(&sum_mutex, NULL);
41      for (int i = 0; i < NUM_THREADS; i++)
42      {
43          thread_data[i].start_index = i * segment_size;
44          thread_data[i].end_index = (i == NUM_THREADS - 1) ? N : (i + 1) * segment_size;
45          pthread_create(&threads[i], NULL, compute_sum, (void *)&thread_data[i]);
46      }
47      for (int i = 0; i < NUM_THREADS; i++)
48      {
49          pthread_join(threads[i], NULL);
50      }
51      printf("Sum of array elements: %d\n", sum);
52      pthread_mutex_destroy(&sum_mutex);
53      return 0;
54  }
```

Figure 5: A concurrent program that sums all the elements in an array using 4 threads under the pthreads framework. The programmer must control the task dispatching (lines 43-45) and data synchronization (lines 25-27) manually.

## B.2 PTHREADS

Pthreads, or POSIX threads, is a low-level thread management library defined by the POSIX standard[3]. Unlike OpenMP, which abstracts many details of parallelism, pthreads provides explicit control over thread creation, synchronization, and resource management, making it suitable for more complex or specialized concurrency requirements.

**Programming model**. Pthreads operates on a **thread-level** model, where threads are explicitly created and managed by the developer. Threads can execute concurrently, sharing memory space, but it is the programmer's responsibility to ensure proper synchronization to avoid issues like data races and deadlocks.

**Fine-grained but manual parallelism management**. Pthreads provides fine-grained control over thread behavior, such as thread priorities, scheduling policies, and thread synchronization mechanisms like mutexes, condition variables, and barriers. This flexibility is essential for low-level system

---

[3]https://posix.opengroup.org/

Table 4: Additional statistics of DRDBench: the 226 programs that contain data races

| Category | N | Origin | Lines of Code | | | Number of Races | | |
|---|---|---|---|---|---|---|---|---|
| | | | Min | Max | Average | Min | Max | Average |
| goblint-regression | 56 | Goblint | 14 | 107 | 35.09 | 1 | 4 | 1.14 |
| ldv-races | 8 | Linux drivers | 65 | 150 | 103.13 | 2 | 7 | 3.5 |
| pthread | 20 | handcrafted | 42 | 140 | 69.45 | 1 | 4 | 2.00 |
| pthread-atomic | 10 | handcrafted | 47 | 322 | 120.90 | 2 | 30 | 12.30 |
| pthread-C-DAC | 1 | C-DAC | 61 | 61 | 61.00 | 2 | 2 | 2.00 |
| pthread-complex | 2 | handcrafted | 260 | 387 | 323.50 | 5 | 11 | 8.00 |
| pthread-deagle | 20 | Deagle | 28 | 55 | 39.25 | 1 | 2 | 1.75 |
| pthread-divine | 8 | DIVINE | 32 | 151 | 90.50 | 1 | 6 | 3.50 |
| pthread-driver-races | 4 | Linux drivers | 509 | 616 | 589.25 | 2 | 7 | 3.25 |
| pthread-ext | 44 | CProver | 33 | 246 | 103.34 | 1 | 30 | 2.32 |
| pthread-lit | 9 | handcrafted | 28 | 125 | 54.56 | 1 | 9 | 3.33 |
| pthread-nondet | 6 | handcrafted | 54 | 62 | 58.33 | 1 | 3 | 2.33 |
| pthread-race-challenges | 37 | Goblint | 25 | 79 | 45.84 | 1 | 8 | 1.41 |
| weaver | 1 | handcrafted | 88 | 88 | 88.00 | 2 | 2 | 2.00 |
| Summary | 226 | | 14 | 616 | 75.81 | 1 | 30 | 2.43 |

programming, real-time applications, or performance-critical systems that require precise control over concurrency. However, such parallelism management relies **entirely on human control**.

**High programming complexity**. While providing powerful tools for concurrency, pthreads increases the complexity of parallel programming. The developer must explicitly manage thread lifecycle, synchronization, and resource sharing by **calling pthreads APIs**. If the API calls are not handled correctly, they can lead to potential errors. There exist many actual cases in pthreads-based programs where improper parallelism management leads to concurrency-related bugs, including the famous DirtyCow bug[4] in the Linux kernel.

**Use Cases**. Pthreads is commonly used in system-level programming, operating systems, networking services, and other low-level applications where detailed control over threading and resource management is required. It is particularly valuable in environments where shared-memory systems need explicit thread control, such as in embedded systems, database engines, or real-time systems.

## C  ADDITIONAL STATISTICS AND PROGRAM EXAMPLES OF DRDBENCH

Tables 4 and 5 present additional statistics for the programs in DRDBench. The `category` indicates the original category within the SV-Benchmarks, `N` refers to the number of programs in that category, and `origin` specifies the source of these programs. Table 6 presents statistics for the patterns of data races within DRDBench.

To demonstrate the variety of programs and data races within DRDBench, we present two program examples in Figures 6 and 7 and illustrate their data races in the corresponding captions.

## D  THE IMPACT OF USER-DEFINED HEADER INLINING

In the construction of DRDBench, we propose flattening multi-file C programs into single-file versions to simplify the evaluation steps. Since many real-world programs are organized in a multi-file structure, a common concern is that this flattening might affect the performance of the evaluated models, potentially leading to biased evaluation results. To alleviate this concern, we conducted a comparative study assessing model performance using both the single-file and the original multi-file versions of the programs.

We randomly selected 40 programs from the DRDBench, including 20 that contain data races and 20 that are free of data races. Each program originally consists of multiple files. To ensure a fair

---

[4] https://dirtycow.ninja/

Table 5: Additional statistics of DRDBench: the 777 programs that are free of data races

| Category | N | Origin | Lines of Code | | |
|---|---|---|---|---|---|
| | | | Min | Max | Average |
| goblint-regression | 205 | Goblint | 15 | 170 | 66.70 |
| ldv-races | 19 | Linux drivers | 64 | 148 | 108.55 |
| pthread | 61 | handcrafted | 42 | 157 | 87.07 |
| pthread-atomic | 18 | handcrafted | 56 | 182 | 103.25 |
| pthread-C-DAC | 5 | C-DAC | 62 | 127 | 94.75 |
| pthread-complex | 6 | handcrafted | 156 | 391 | 275.75 |
| pthread-deagle | 24 | Deagle | 58 | 89 | 73.00 |
| pthread-divine | 10 | DIVINE | 32 | 39 | 35.5 |
| pthread-driver-races | 22 | Linux drivers | 509 | 624 | 601.00 |
| pthread-ext | 95 | CProver | 36 | 246 | 97.80 |
| pthread-lit | 14 | handcrafted | 40 | 85 | 55.40 |
| pthread-race-challenges | 63 | Goblint | 25 | 79 | 46.58 |
| pthread-wmm | 283 | handcrafted | 118 | 427 | 247.08 |
| weaver | 172 | handcrafted | 55 | 466 | 113.06 |
| Summary | 777 | | 15 | 624 | 159.49 |

```
01  #include <pthread.h>
02  #include <assert.h>
03
04  pthread_mutex_t lock;
05  pthread_cond_t cond;
06  int x;
07  bool x_set = 0;
08
09  void *thread(void *arg)
10  {
11      (void)arg;
12      pthread_mutex_lock(&lock);
13      while (!x_set)
14          pthread_cond_wait(&cond, &lock);
15      assert(x == 42);
16      pthread_mutex_unlock(&lock);
17      return NULL;
18  }
19
20  int main()
21  {
22      pthread_t t;
23      pthread_create(&t, NULL, thread, NULL);
24      for (int i = 0; i <= 42; i++)
25          x = i;
26      x_set = 1;
27      pthread_cond_broadcast(&cond);
28      pthread_join(t, NULL);
29  }
```

Figure 6: The program contains a trivial data race on the variable x_set between lines 13 and 26. Interestingly, the variable x does not experience a data race, as the signal/wait mechanism and the while loop at line 13 ensure that the two accesses to x at lines 15 and 25 cannot run concurrently.

```
01  #include <stdlib.h>
02  #include <pthread.h>
03  #include <semaphore.h>
04  int data = 0;
05  sem_t data_sem;
06  void assume_abort_if_not(int cond)
07  {
08      if (!cond)
09      {
10          abort();
11      }
12  }
13
14  void *thread(void *arg)
15  {
16      sem_wait(&data_sem);
17      data = __VERIFIER_nondet_int();
18      sem_post(&data_sem);
19      return NULL;
20  }
21
22  int main()
23  {
24      sem_init(&data_sem, 0, 2);
25      int threads_total = __VERIFIER_nondet_int();
26      assume_abort_if_not(threads_total >= 0);
27      pthread_t *tids = malloc(threads_total * sizeof(pthread_t));
28      for (int i = 0; i < threads_total; i++)
29      {
30          pthread_create(&tids[i], NULL, &thread, NULL);
31      }
32      for (int i = 0; i < threads_total; i++)
33      {
34          pthread_join(tids[i], NULL);
35      }
36      free(tids);
37      return 0;
38  }
```

Figure 7: The program contains a data race on the variable `data` across multiple threads that concurrently reach line 17. Interestingly, the semaphore `data_sem` does not eliminate the data race, as it is initialized with a value of 2 (line 24), allowing at most two threads to enter the critical section (lines 16–18) simultaneously.

Table 6: Concurrency patterns of programs within DRDBench.

| Category | Race Patterns | | Number of Threads | | | Number of Shared Variables | | |
|---|---|---|---|---|---|---|---|---|
| | Read-Write | Write-Write | Min | Max | Average | Min | Max | Average |
| goblint-regression | 22 | 42 | 2 | 30,000 | 3,221.06 | 1 | 300 | 4.31 |
| ldv-races | 10 | 18 | 2 | 2 | 2.00 | 1 | 2 | 1.68 |
| pthread-atomic | 99 | 24 | 2 | 4 | 2.89 | 2 | 7 | 4.06 |
| pthread-C-DAC | 1 | 1 | 2 | 8 | 3.20 | 1 | 5 | 2.80 |
| pthread-deagle | 35 | 0 | 2 | 51 | 19.29 | 1 | 8 | 2.58 |
| pthread-divine | 27 | 1 | 2 | 2 | 2.00 | 1 | 4 | 1.90 |
| pthread-ext | 88 | 14 | 2 | 321 | 9.11 | 1 | 6 | 3.53 |
| pthread-lit | 24 | 6 | 2 | 51 | 6.14 | 1 | 4 | 3.07 |
| pthread-nondet | 8 | 6 | 9 | 21 | 13.67 | 2 | 2 | 2.00 |
| pthread-race-challenges | 30 | 22 | 2 | 5 | 2.40 | 1 | 9 | 2.49 |
| pthread-wmm | 0 | 0 | 2 | 4 | 3.15 | 3 | 45 | 18.65 |
| pthread | 34 | 6 | 2 | 16 | 2.89 | 1 | 5 | 2.39 |
| weaver | 0 | 2 | 2 | 8 | 2.98 | 1 | 23 | 6.94 |
| pthread-complex | 9 | 7 | 2 | 7 | 4.17 | 1 | 11 | 6.83 |
| pthread-driver-races | 13 | 0 | 2 | 2 | 2.00 | 3 | 13 | 6.64 |
| Summary | 400 | 149 | 2.47 | 2,033.47 | 219.80 | 1.40 | 29.60 | 4.66 |

Table 7: The impact of user-defined header inlining.

| Model | Pass@1 | Recall | Precision | F1 | 1-FPR | Average |
|---|---|---|---|---|---|---|
| DeepSeek-R1-671B | 85.00% | 89.36% | 97.67% | 93.33% | 80.00% | 89.07% |
| w/ inlining | 85.00% | 93.62% | 97.78% | 95.65% | 95.00% | 93.41% |
| Diff (relative) | 0.00% | ↑4.77% | ↑0.11% | ↑2.49% | ↑18.75% | ↑4.87% |
| Qwen-QwQ | 70.00% | 82.98% | 92.86% | 87.64% | 80.00% | 82.70% |
| w/ inlining | 75.00% | 85.11% | 85.11% | 85.11% | 90.00% | 84.07% |
| Diff (relative) | ↑7.14% | ↑2.57% | ↓8.35% | ↓2.89% | ↑12.50% | ↑1.66% |
| Qwen3-Thinking-32B | 65.00% | 72.34% | 77.27% | 74.73% | 75.00% | 72.87% |
| w/ inlining | 65.00% | 79.72% | 86.05% | 82.22% | 95.00% | 81.60% |
| Diff (relative) | 0.00% | ↑10.20% | ↑11.36% | ↑10.02% | ↑26.67% | ↑11.98% |
| DeepSeek-V3-671B | 65.00% | 74.47% | 83.33% | 78.65% | 50.00% | 70.29% |
| w/ inlining | 60.00% | 65.96% | 79.49% | 72.09% | 60.00% | 67.51% |
| Diff (relative) | ↓7.69% | ↓11.43% | ↓4.61% | ↓8.34% | ↑20.00% | ↓3.96% |
| Qwen2.5-72B | 5.00% | 19.15% | 42.86% | 26.47% | 75.00% | 33.70% |
| w/ inlining | 0.00% | 21.28% | 30.30% | 25.00% | 85.00% | 32.32% |
| Diff (relative) | ↓100.00% | ↑11.12% | ↓29.30% | ↓5.55% | ↑13.33% | ↓4.10% |
| Qwen2.5-Coder-32B | 20.00% | 38.30% | 64.29% | 48.00% | 70.00% | 48.12% |
| w/ inlining | 30.00% | 46.81% | 66.67% | 55.00% | 75.00% | 54.70% |
| Diff (relative) | ↑50.00% | ↑22.22% | ↑3.70% | ↑14.58% | ↑7.14% | ↑13.67% |

evaluation, we reused the same prompt described in Appendix H. To tackle the challenge of line number identification in multi-file programs, we labeled the line numbers of each file in a continuous format. For example, if a program had two files with 10 and 20 lines respectively, the lines in the first file were labeled from 1 to 10, while the lines in the second file were labeled from 11 to 30. We also included the filename before the content of each file in the prompt.

We evaluated the top three best-performing reasoning models (DeepSeek-R1, Qwen-QwQ, and Qwen3-Thinking-32B), alongside the top three best-performing non-reasoning models (DeepSeek-V3, Qwen2.5-72B, and Qwen2.5-Coder-32B) selected from the main experiment. We assessed these six models once on the 40 programs using the same metrics as the main evaluation: pass@1, recall, precision, F1, and FPR. To align the direction of all metrics for easier comparison, we transformed the FPR metric to $1 - $ FPR (since FPR is smaller-is-better while all others are larger-is-better). We calculated an average score across all metrics (pass@1, recall, precision, F1, and $1 - $ FPR) to provide a general overview of the performance divergences between single-file and multi-file settings. Finally, we calculated the relative performance difference for each metric to precisely quantify the degree of improvement or degradation across the two settings.

Table 8: The impact of comment removal.

| Model | Pass@1 | Recall | Precision | F1 | 1-FPR | Average |
|---|---|---|---|---|---|---|
| DeepSeek-R1-671B | 65.00% | 82.35% | 57.14% | 67.47% | 95.00% | 73.39% |
| w/ comment removal | 70.00% | 83.82% | 70.37% | 76.51% | 90.00% | 78.14% |
| Diff (relative) | ↑7.69% | ↑1.79% | ↑23.15% | ↑13.40% | ↓5.26% | ↑6.47% |
| Qwen-QwQ | 50.00% | 57.35% | 62.90% | 60.00% | 85.00% | 63.05% |
| w/ comment removal | 60.00% | 58.82% | 72.73% | 65.04% | 90.00% | 69.32% |
| Diff (relative) | ↑20.00% | ↑2.56% | ↑15.63% | ↑8.40% | ↑5.88% | ↑9.94% |
| Qwen3-Thinking-32B | 35.00% | 35.29% | 57.14% | 43.64% | 85.00% | 51.21% |
| w/ comment removal | 40.00% | 41.18% | 65.12% | 50.45% | 70.00% | 53.35% |
| Diff (relative) | ↑14.29% | ↑16.69% | ↑13.97% | ↑15.60% | ↓17.65% | ↑4.17% |
| DeepSeek-V3-671B | 35.00% | 44.12% | 49.18% | 46.51% | 85.00% | 51.96% |
| w/ comment removal | 35.00% | 63.24% | 38.39% | 47.78% | 45.00% | 45.88% |
| Diff (relative) | 0.00% | ↑43.34% | ↓21.94% | ↑2.73% | ↓47.06% | ↓11.70% |
| Qwen2.5-72B | 10.00% | 16.18% | 20.00% | 17.89% | 85.00% | 29.81% |
| w/ comment removal | 10.00% | 17.65% | 32.43% | 22.86% | 80.00% | 32.59% |
| Diff (relative) | 0.00% | ↑9.09% | ↑62.15% | ↑27.78% | ↓5.88% | ↑9.30% |
| Qwen2.5-Coder-32B | 35.00% | 20.59% | 28.57% | 23.93% | 75.00% | 36.62% |
| w/ comment removal | 25.00% | 20.59% | 35.90% | 26.17% | 75.00% | 36.53% |
| Diff (relative) | ↓28.57% | 0.00% | ↑25.66% | ↑9.36% | 0.00% | ↓0.23% |

The experimental results are presented in Table 7. In this table, we use green to highlight scores where the model performs better with single-file programs, while red indicates the opposite scenario. Overall, the experimental results show that flattening multi-file programs into single-file versions has a limited impact on the model's race detection performance: the relative performance difference is typically within $5\%$ and not exceeding $15\%$ when evaluated by the `average` score across all metrics. We do observe several substantial changes in the `pass@1` metric, including a $100\%$ degradation and a $50\%$ improvement. However, we attribute these extreme percentage fluctuations to small base numbers in the relative difference calculation, confirming they do not indicate a fundamental change in overall model performance. Furthermore, we find that the $1-\text{FPR}$ metric is significantly improved when applying the flattening operation. We hypothesize that this improvement occurs because the file flattening reduces the overall context length, making the program structure and logic more accessible for the models to comprehend.

# E THE IMPACT OF COMMENT REMOVAL

In the construction of DRDBench, we propose removing the code comments from the source code. This is necessary to ensure that LLMs must rely solely on their internal understanding of the code to conduct race detection, preventing them from exploiting human intent or external explanations encoded as comments. Although this step is crucial for evaluating LLMs' comprehension of concurrent programming logic, it may bias the evaluation results when the objective is to assess performance in real-world settings. To investigate the practical influence of this comment removal operation, we conducted a comparative study assessing model performance on programs both with and without comments.

We randomly selected 40 programs from the DRDBench, including 20 that contain data races and 20 that are free of data races. These 40 programs can be different from those evaluated in Appendix D because this experiment does not require the evaluated programs to consist of multiple files. To ensure a fair evaluation, we reused the same prompt described in Appendix H. Following the evaluation settings described in Appendix D, we evaluated the top three best-performing reasoning models (DeepSeek-R1, Qwen-QwQ, and Qwen3-Thinking-32B), alongside the top three best-performing non-reasoning models (DeepSeek-V3, Qwen2.5-72B, and Qwen2.5-Coder-32B) selected from the main experiment. We assessed these six models once on the 40 programs using the same metrics as Appendix D: `pass@1`, `recall`, `precision`, `F1`, `1 - FPR`, and `average`.

Table 9: The impact of code formatting.

| Model | Pass@1 | Recall | Precision | F1 | 1-FPR | Average |
|---|---|---|---|---|---|---|
| DeepSeek-R1-671B | 70.00% | 79.41% | 62.07% | 69.68% | 100.00% | 76.23% |
| w/ formatting | 70.00% | 83.82% | 70.37% | 76.51% | 90.00% | 78.14% |
| Diff (relative) | 0.00% | ↑5.55% | ↑13.37% | ↑9.80% | ↓10.00% | ↑2.50% |
| Qwen-QwQ | 40.00% | 61.76% | 70.00% | 65.63% | 65.00% | 60.48% |
| w/ formatting | 60.00% | 58.82% | 72.73% | 65.04% | 90.00% | 69.32% |
| Diff (relative) | ↑50.00% | ↓4.76% | ↑3.90% | ↓0.90% | ↑38.46% | ↑14.62% |
| Qwen3-Thinking-32B | 35.00% | 42.65% | 64.44% | 51.33% | 70.00% | 52.68% |
| w/ formatting | 40.00% | 41.18% | 65.12% | 50.45% | 70.00% | 53.35% |
| Diff (relative) | ↑14.29% | ↓3.45% | ↑1.06% | ↓1.71% | 0.00% | ↑1.26% |
| DeepSeek-V3-671B | 30.00% | 44.12% | 42.25% | 43.17% | 75.00% | 46.91% |
| w/ formatting | 35.00% | 63.24% | 38.39% | 47.78% | 45.00% | 45.88% |
| Diff (relative) | ↑16.67% | ↑43.34% | ↓9.14% | ↑10.68% | ↓40.00% | ↓2.19% |
| Qwen2.5-72B | 10.00% | 13.24% | 20.45% | 16.07% | 80.00% | 27.95% |
| w/ formatting | 10.00% | 17.65% | 32.43% | 22.86% | 80.00% | 32.59% |
| Diff (relative) | 0.00% | ↑33.31% | ↑58.58% | ↑42.25% | 0.00% | ↑16.59% |
| Qwen2.5-Coder-32B | 25.00% | 19.12% | 48.15% | 27.37% | 75.00% | 38.93% |
| w/ formatting | 25.00% | 20.59% | 35.90% | 26.17% | 75.00% | 36.53% |
| Diff (relative) | 0.00% | ↑7.69% | ↓25.44% | ↓4.38% | 0.00% | ↓6.15% |

The experimental results are presented in Table 8. In this table, we use green to highlight scores where the model performs better after removing the comments, while red indicates the opposite scenario. Overall, the results demonstrate that removing the code comments has a limited impact on the model's race detection performance: the relative performance improvement or degradation typically does not exceed $10\%$ when evaluated by the `average` metric. Consequently, we conclude that this preprocessing method does not significantly bias the evaluation results. Interestingly, we observe a divergence in metric performance: retaining comments helps LLMs reduce their false positive reports on race-free programs (improving the $1-$ `FPR` metric), yet this retention conversely leads to an increase in false positive reports on programs containing races. We hypothesize that this divergence occurs because race-free programs can contain code comments that encode a high-level description of the synchronization mechanisms (i.e., what is constrained from running concurrently). LLMs are able to exploit this information to correctly confirm the absence of a race, thereby reducing false positives. Conversely, programs containing data races usually possess fewer synchronization mechanisms and, consequently, fewer such protective code comments for LLMs to exploit. Critically, the presence of the other concurrency-irrelevant comments may degrade the reasoning process, leading to a corresponding decrease in both `recall` and `precision` on these complex programs.

# F  THE IMPACT OF CODE FORMATTING

In the construction of DRDBench, we utilize the tool `clang-format` to standardize the coding style of the programs. This standardization is necessary to ensure that line numbers remain a sufficient mechanism for precisely representing the race locations. However, this formatting operation carries the potential risk of inadvertently simplifying the race detection task by altering the original code structure. Although we manually inspected the formatted code to confirm that the programs function identically before and after the code formatting, we conducted a comparative study to rigorously assess its influence on model performance.

We randomly selected 40 programs from the DRDBench, including 20 that contain data races and 20 that are free of data races. These 40 programs are the same as those evaluated in Appendix E. To ensure a fair evaluation, we reused the same prompt described in Appendix H. Following the evaluation settings described in Appendix D, we evaluated the top three best-performing reasoning models (DeepSeek-R1, Qwen-QwQ, and Qwen3-Thinking-32B), alongside the top three best-performing non-reasoning models (DeepSeek-V3, Qwen2.5-72B, and Qwen2.5-Coder-32B) selected from the

Table 10: The combined impact of three preprocessing methods.

| Model | Pass@1 | Recall | Precision | F1 | 1-FPR | Average |
|-------|--------|--------|-----------|-----|-------|---------|
| DeepSeek-R1-671B | 59.09% | 75.34% | 60.44% | 67.07% | 83.33% | 69.05% |
| w/ preprocessing | 68.18% | 73.97% | 72.00% | 72.97% | 91.03% | 75.63% |
| Diff (relative) | ↑15.38% | ↓1.82% | ↑19.13% | ↑8.80% | ↑9.24% | ↑9.52% |
| Qwen-QwQ | 40.91% | 43.84% | 66.67% | 52.89% | 80.77% | 57.02% |
| w/ preprocessing | 59.09% | 60.27% | 73.33% | 66.17% | 89.74% | 69.72% |
| Diff (relative) | ↑44.44% | ↑37.48% | ↑9.99% | ↑25.11% | ↑11.11% | ↑22.28% |
| Qwen3-Thinking-32B | 31.82% | 30.14% | 52.38% | 38.26% | 84.62% | 47.44% |
| w/ preprocessing | 36.36% | 39.73% | 65.91% | 49.57% | 80.77% | 54.47% |
| Diff (relative) | ↑14.27% | ↑31.82% | ↑25.83% | ↑29.56% | ↓4.55% | ↑14.80% |
| DeepSeek-V3-671B | 31.82% | 47.94% | 50.00% | 48.95% | 75.64% | 50.87% |
| w/ preprocessing | 36.36% | 60.27% | 38.94% | 47.31% | 50.00% | 46.58% |
| Diff (relative) | ↑14.27% | ↑25.72% | ↓22.12% | ↓3.35% | ↓33.90% | ↓8.44% |
| Qwen2.5-72B | 13.64% | 24.66% | 36.73% | 29.51% | 84.62% | 37.83% |
| w/ preprocessing | 13.64% | 20.55% | 42.86% | 27.78% | 83.33% | 37.63% |
| Diff (relative) | 0.00% | ↓16.67% | ↑16.69% | ↓5.86% | ↓1.52% | ↓0.53% |
| Qwen2.5-Coder-32B | 22.73% | 20.55% | 34.88% | 25.86% | 69.23% | 34.65% |
| w/ preprocessing | 22.73% | 21.92% | 30.77% | 25.60% | 70.51% | 34.31% |
| Diff (relative) | 0.00% | ↑6.67% | ↓11.78% | ↓1.01% | ↑1.85% | ↓0.99% |

main experiment. We assessed these six models once on the 40 programs using the same metrics as Appendix D: `pass@1`, `recall`, `precision`, F1, `1 - FPR`, and `average`.

The experimental results are presented in Table 9. In this table, we use green to highlight scores where the model performs better after formatting, while red indicates the opposite scenario. Overall, the results demonstrate that standardizing the coding styles typically has a limited impact on the model's race detection performance: the relative performance improvement or degradation typically does not exceed **15**% when evaluated by the `average` metric. Consequently, we conclude that this preprocessing method does not significantly bias the evaluation results. Furthermore, the influence of code formatting is highly model-dependent; we observed no uniform or generalizable trend regarding performance changes across the entire set of evaluated models in this experiment.

## G   THE COMBINED IMPACT OF THREE PREPROCESSING METHODS

To rigorously evaluate the combined influence of the three preprocessing methods and assess the evaluation bias they may introduce compared to a real-world setting, we conducted a comparative experiment. Specifically, we compared the performance of LLMs on both the preprocessed programs and the originally formatted programs. To conserve computational resources, this evaluation was performed on a random 10% split of DRDBench. This reduced split consists of 100 programs, comprising 22 programs containing data races and 78 race-free programs. We assessed a total of six models: the top three best-performing reasoning models (DeepSeek-R1, Qwen-QwQ, and Qwen3-Thinking-32B) and the top three best-performing non-reasoning models (DeepSeek-V3, Qwen2.5-72B, and Qwen2.5-Coder-32B) selected from the main experiment. We evaluated these six models once on the 100 programs using the same suite of metrics defined in Appendix D: `pass@1`, `recall`, `precision`, F1, `1 - FPR`, and the `average`.

The experimental results for the combined preprocessing influence are presented in Table 10. In this table, we use green to highlight scores where the model performs better after preprocessing, while red indicates the opposite scenario (performance degradation). Overall, the results demonstrate that preprocessing typically has a minimal impact on non-reasoning models, but it can significantly improve the performance of reasoning LLMs. We hypothesize that this divergence exists because reasoning LLMs are often specifically trained to optimize their reasoning capability, potentially at the expense of handling real-world code complexities, such as multi-file structures and varied coding styles. Thus, the preprocessing steps help simplify these code patterns, enabling the reasoning models to achieve superior performance. Crucially, it is essential to note that this observed bias does

Table 11: The impact of domain-specific knowledge in the prompt.

| Model | Pass@1 | Recall | Precision | F1 | 1-FPR | Average |
|---|---|---|---|---|---|---|
| DeepSeek-R1-671B | 65.49% | 90.71% | 53.78% | 66.85% | 87.90% | 72.95% |
| +DK | 68.14% | 75.23% | 75.36% | 75.30% | 86.87% | 76.18% |
| Diff (relative) | ↑4.05% | ↓17.07% | ↑40.13% | ↑12.64% | ↓1.17% | ↑4.43% |
| Qwen-QwQ | 52.65% | 57.01% | 64.54% | 60.15% | 50.58% | 56.99% |
| +DK | 60.62% | 65.03% | 77.61% | 70.76% | 87.39% | 72.28% |
| Diff (relative) | ↑15.14% | ↑14.07% | ↑20.25% | ↑17.64% | ↑72.78% | ↑26.84% |
| Qwen3-Thinking-32B | 43.36% | 52.09% | 62.04% | 56.63% | 90.35% | 60.89% |
| +DK | 46.90% | 48.82% | 73.22% | 58.58% | 87.26% | 62.96% |
| Diff (relative) | ↑8.16% | ↓6.28% | ↑18.02% | ↑3.44% | ↓3.42% | ↑3.39% |
| DeepSeek-V3-671B | 38.05% | 51.91% | 42.73% | 46.71% | 59.59% | 47.80% |
| +DK | 50.88% | 55.19% | 54.69% | 54.94% | 50.19% | 53.18% |
| Diff (relative) | ↑33.72% | ↑6.32% | ↑27.99% | ↑17.62% | ↓15.77% | ↑11.26% |
| Qwen2.5-72B | 16.81% | 17.85% | 62.03% | 27.72% | 72.72% | 39.43% |
| +DK | 28.32% | 33.70% | 56.23% | 42.14% | 84.07% | 48.89% |
| Diff (relative) | ↑68.47% | ↑88.80% | ↓9.35% | ↑52.02% | ↑15.61% | ↑24.01% |
| Qwen2.5-Coder-32B | 24.36% | 18.94% | 48.83% | 27.30% | 61.52% | 36.19% |
| +DK | 27.88% | 28.60% | 50.81% | 36.60% | 72.07% | 43.19% |
| Diff (relative) | ↑14.45% | ↑51.00% | ↑4.05% | ↑34.07% | ↑17.15% | ↑19.35% |

not overturn the core insights obtained in our main evaluation (Section 5), as the reasoning models still significantly outperform the non-reasoning ones even when the preprocessing methods are not applied. Therefore, the actionable insights proposed in Section 5 remain valid.

## H  PROMPT TEMPLATE FOR THE FINEEVAL-RACE

Listing 1 presents the prompt template used in our fine-grained evaluation framework, FineEval-Race. Note that the blue lines are only for illustration purposes. They are not a part of the prompt template.

## I  THE IMPACT OF DOMAIN-SPECIFIC KNOWLEDGE IN THE PROMPT

Exploring the impact of domain knowledge in the prompt could provide valuable insights, e.g., whether it is necessary to include such information in the prompt and whether the models have learnt such knowledge during the pre-training. To this end, we conducted an additional experiment. We utilized the top three best-performing reasoning models (DeepSeek-R1, Qwen-QwQ, and Qwen3-Thinking-32B) and non-reasoning models (DeepSeek-V3, Qwen2.5-72B, and Qwen2.5-Coder-32B) from our main experiment to conduct this experiment. In this experiment, we removed the *Domain-specific knowledge introduction* section from the prompt and compared the model's performance with the complete prompt. We evaluated these six models using the same suite of metrics defined in Appendix D: `pass@1`, `recall`, `precision`, `F1`, `1 - FPR`, and the `average`.

The experimental results are presented in Table 11, where DK stands for domain knowledge. We use green to highlight the scores where introducing the domain knowledge brings improvements, while red indicates the opposite scenario. Although the models did not achieve zero scores when no domain knowledge was provided—indicating they had acquired some latent level of domain knowledge during pre-training—we found that incorporating explicit domain knowledge significantly improves their overall performance. When evaluated by the `average` metric, the relative performance improvement is **15**% on average and **27**% at maximum. We attribute this substantial enhancement to the incomplete nature of the domain knowledge acquired during pre-training. To ensure a fair comparison and to eliminate performance differences stemming from varied pre-training processes, we designed the prompt to include a dedicated section for domain knowledge. This methodological choice guarantees that all models have access to a complete and standardized knowledge base, thereby allowing our evaluation to focus specifically on assessing the models' inherent race detection capabilities rather than the completeness of their pre-training knowledge.

Listing 1: Prompt template of FineEval-Race

## Role and task definition
You are an expert at concurrent program design and data race detection. In the following, you will be given a program. You'll need to carefully look over the program to check whether it contains data race bugs. If it contains data race bugs, please locate them in line number pairs.
## Domain-specific knowledge introduction
The data race bug is a bug that occurs when (1) two or more threads access a shared variable at the same time, and (2) at least one of the accesses is a write. Note that, two operations **cannot** execute at the same time when (1) both are atomical operations, (2) both are protected by the same mutex, (3) they are guarded by a semaphare which ensures the exclusive access of the shared variable, or (4) other mechanism that forbids the two operations to execute at the same time.
The program can use `__VERIFIER_atomic_begin()` and `__VERIFIER_atomic_end()` to mark the start and the end of an atomic zone. Besides, if the function name has the `__VERIFIER_atomic` prefix, the corresponding function should also be regarded as an atomic zone. All operations inside the atomic zone should be regarded as atomic.
The program can use `pthread_mutex_lock(&m)` and `pthread_mutex_unlock(&m)` to lock and unlock a mutex `m`.
The program can use `sem_wait()` and `sem_post()` to control semaphores; they do not lock or unlock mutexes. A semaphore holds an integer value. The `sem_wait()` is used to decrease the semaphore's value (typically by 1) to signal that the program wants to enter a critical section or use a resource. If the semaphore's value is greater than 0, `sem_wait()` decrements it and then proceeds. If the semaphore's value is 0, `sem_wait()` is blocked until the semaphore's value becomes greater than 0. The `sem_post` is used to increment the semaphore's value (typically by 1), indicating that a resource has been released.
The program can use `pthread_create()` to create a new thread and use `pthread_join()` to join the created thread. All the operations inside the new thread should happen after the `pthread_create()` site and before the `pthread_join()` site.
The program can use `pthread_cond_wait()` and `pthread_cond_signal()` to wait and signal a condition variable. It can also use `pthread_barrier_wait()` to wait for a barrier.
The program also uses `assume_abort_if_not()` as `assert()`. It can use `__VERIFIER_nondet_int()` to get a random integer. Besides, the indices of the lines are provided at the beginning of each line, e.g., "1:", to help locate the line numbers.
## Step-by-step description of the detection procedure
You can follow the following steps to detect the data race bugs:
1. Read the program carefully and understand how the threads are created and joined.
2. Check the shared variables and their accesses.
3. Check the synchronization mechanisms (atomic zones, mutexes, semaphores, condition variables, etc.) and their usage.
4. For each pair of accesses to the same shared variable, check whether they can constitute a data race.
## Output format instructions
After thoroughly checking all potential data race bugs, please output all the confirmed data races. If no data race is found, please answer with an empty list. Please answer in the following JSON format (each race as one dict):
```json
{
    "races": [
        {
            "shared_variable": "the name of the same shared variable",
            "lineA": the line number of the first access in `int` format,
            "lineB": the line number of the second access in `int` format
        },
    ...]
}```
## Source code of the program
<The code to be analyzed, with each line prepended by its corresponding line number.>

## J  TRADE-OFF BETWEEN FALSE POSITIVES AND FALSE NEGATIVES

In the evaluation, we propose to solely check whether the line number matches the ground-truth results to decide the correctness of the model's answers. We acknowledge that this simplified evaluation method may introduce false positives in extreme cases. For instance, consider a scenario where two variables ($x$ and $y$) appear on the same two lines of code. If only the variable $x$ is involved in a data race, the ground truth should point solely to $x$. When an LLM incorrectly reports a data race on $y$, our framework can erroneously treat it as a true positive, as the line numbers match. This can happen when the variable $y$ is only read on both lines; thus, its two accesses cannot constitute a data race. It is essential to note that if the variable $y$ is written on either line, its race status should remain consistent with that of the variable $x$ because the synchronization within a given code line does not change. In such cases, the line-level comparison will not introduce a false positive.

However, we argue that such false positives are not common in practice. This is substantiated by our observation that, having explicitly provided the LLMs with the precise definition of a data race, which requires at least one operation to be a write, the models do not report two read operations to constitute a data race.

Furthermore, we are concerned that incorporating variable names into the evaluation can introduce a significant risk of false negatives. This is because variable aliasing and array indexing can result in the same memory object being referenced by different names. Since it is challenging for human annotators to exhaustively identify all possible name variants for every racing variable, any misalignment between the human-annotated ground truth and the names generated by the LLMs would erroneously classify true positive results as false negatives. Moreover, given that different LLMs may exhibit different naming preferences, a race examination based on the conjunction of line number and variable name may introduce evaluation bias, unfairly favoring models whose naming conventions align with those of the human annotators. Considering this challenge, we propose to only check the line numbers in our evaluation for a fairer comparison between different LLMs.

## K  DECODING HYPERPARAMETERS

Table 12: Default decoding hyperparameters of LLMs

| Model Series | Temperature | Top_p | Top_k |
| --- | --- | --- | --- |
| DeepSeek-R1 | 0.6 | 0.95 | N/A(-1) |
| DeepSeek-V3 | N/A(1.0) | N/A(1.0) | N/A(-1) |
| R1-Distill-Qwen2.5 | 0.6 | 0.95 | N/A(-1) |
| R1-Distill-Llama | 0.6 | 0.95 | N/A(-1) |
| Qwen-QwQ | 0.6 | 0.95 | 40 |
| Qwen-3 | 0.6 | 0.95 | 20 |
| Qwen-2.5 | 0.7 | 0.8 | 20 |
| Qwen-2.5-Coder | 0.7 | 0.8 | 20 |
| Llama-3.1 | 0.6 | 0.9 | N/A(-1) |

In the experiment, we use the default decoding hyperparameters recommended by each model's source code for multi-sampling. These default hyperparameters are summarized in Table 12. An entry of "N/A" indicates that the source code does not specify a default value for the hyperparameter. In such cases, we set temperature=1.0, top_p=1.0, and top_k=-1. The top_k=-1 means that the top-K sampling mechanism is disabled.

## L  FINANCIAL COST OF EVALUATING CLOSED-SOURCE COMMERCIAL MODELS

In our experiments, we did not evaluate closed-source commercial models. This was primarily due to the high financial cost associated with closed-source commercial models. We estimated the financial cost of using several closed-source commercial models, including GPT-5, GPT-4o, OpenAI-o1, OpenAI-o3, Claude-3.7 Sonnet, and Claude Opus 4, as detailed in Table 13. For these estimations, we utilized the token consumption information of DeepSeek-V3 (8M prompt tokens + 3M completion tokens) to assess the cost of non-reasoning commercial models. Additionally, we used the data from

Table 13: The estimated financial cost of evaluating closed-source commercial models.

| Model | $ per 1M prompt tokens | $ per 1M completion tokens | Estimated financial cost |
|---|---|---|---|
| DeepSeek-V3 | $0.28 | $0.88 | $4.90 |
| GPT-5 (Nothinking) | $1.25 | $10.00 | $40.00 |
| GPT-4o | $2.50 | $10.00 | $50.00 |
| Claude-3.7 Sonnet (Nothinking) | $3.00 | $15.00 | $69.00 |
| Claude Opus 4 (Nothinking) | $15.00 | $75.00 | $345.00 |
| DeepSeek-R1 | $0.50 | $2.15 | $40.55 |
| GPT-5 (Thinking) | $1.25 | $10.00 | $180.00 |
| OpenAI-o1 | $15.00 | $60.00 | $1140.00 |
| OpenAI-o3 | $2.00 | $8.00 | $152.00 |
| Claude-3.7 Sonnet (Thinking) | $3.00 | $15.00 | $279.00 |
| Claude Opus 4 (Thinking) | $15.00 | $75.00 | $1395.00 |

DeepSeek-R1 (8M prompt tokens + 17M completion tokens) to evaluate the reasoning commercial models. The pricing was sourced from OpenRouter [5].

In conclusion, the use of these closed-source commercial models is quite expensive and will significantly exceed the cost associated with DeepSeek-V3 or DeepSeek-R1. Evaluating these closed-source models will cost hundreds or even thousands of dollars, which surpasses our current budget limit. We plan to assess these models in the future if we can secure additional funding. Furthermore, we intend to make the DRDBench dataset publicly available, enabling other researchers to evaluate these closed-source commercial models using the DRDBench.

## M  THE COMPARISON BETWEEN REASONING AND NON-REASONING MODELS

Table 14 presents a comparison between reasoning models and their non-reasoning counterparts. In this analysis, we focus on the metrics of pass@1, recall, precision, F1, and 1-FPR under greedy decoding. We calculate the average score of all other metrics as average to provide a general overview of the performance divergences. We use green to highlight the scores where reasoning models outperform their non-reasoning counterparts, while red indicates the opposite scenario. The evaluation confirms that reasoning models tend to significantly outperform their non-reasoning counterparts, a trend that is particularly pronounced in models with larger parameter sizes. We observed only one minor exception to this pattern: the Qwen3-NoThinking-1.7B model surpassed its reasoning counterpart by a performance margin of $10.16\%$. However, given the overwhelming trend across all other model sizes and families, we conclude that this isolated exception does not invalidate the core finding regarding the superiority of reasoning capabilities for this task.

## N  THE SOLUTION CHAIN COMPARISON

This section presents a qualitative analysis comparing the solution chains of reasoning and non-reasoning models to demonstrate that the observed performance superiority of reasoning models is primarily introduced by improvements in their underlying reasoning logic. We utilize the solution chains generated by DeepSeek-R1 (the best-performing reasoning model) and DeepSeek-V3 (the best-performing non-reasoning model) from our main evaluation, focusing on the program illustrated in Figure 6 as a representative case study to substantiate this claim.

The solution chains for the case study program are presented in Listings 2 (DeepSeek-R1) and 3 (DeepSeek-V3), respectively. The solution chain of DeepSeek-V3 demonstrates a superficial detection strategy: it identifies data races simply by observing whether the shared variable is protected by a common mutex. This non-reasoning approach fails to integrate code semantics or execution logic, leading directly to the incorrect report of a false positive data race on variable $x$. In sharp contrast, the reasoning model, DeepSeek-R1, initially exhibits the same error, deeming the two accesses to variable $x$ to be a data race. However, it immediately engages in self-correction and reflection (signaled by the internal monologue, "But wait"). DeepSeek-R1 then proceeds to detailly reason about the concrete concurrency semantics surrounding the accesses to $x$ and ultimately determines that the program semantics prevent the two accesses from running concurrently. This case study demonstrates that the

---

[5]https://openrouter.ai/

Table 14: The comparison between reasoning models and their non-reasoning counterparts

| Model | Pass@1 | Recall | Precision | F1 | 1-FPR | Average |
|---|---|---|---|---|---|---|
| DeepSeek-R1-671B | 68.14% | 75.23% | 75.36% | 75.30% | 86.87% | 76.18% |
| DeepSeek-V3-671B | 50.88% | 55.19% | 54.69% | 54.94% | 50.19% | 53.18% |
| Diff (relative) | ↑33.92% | ↑36.31% | ↑37.79% | ↑37.06% | ↑73.08% | ↑43.25% |
| Qwen3-Thinking-32B | 46.90% | 48.82% | 73.22% | 58.58% | 87.26% | 62.96% |
| Qwen3-Nothinking-32B | 19.03% | 27.50% | 18.11% | 21.84% | 86.23% | 34.54% |
| Diff (relative) | ↑146.45% | ↑77.53% | ↑304.31% | ↑168.22% | ↑1.19% | ↑82.26% |
| Qwen3-Thinking-30B-A3B | 43.81% | 45.17% | 69.47% | 54.75% | 80.95% | 58.83% |
| Qwen3-Nothinking-30B-A3B | 16.81% | 29.51% | 12.97% | 18.02% | 15.57% | 18.58% |
| Diff (relative) | ↑160.62% | ↑53.07% | ↑435.62% | ↑203.83% | ↑419.91% | ↑216.70% |
| Qwen3-Thinking-8B | 29.65% | 35.70% | 57.48% | 44.04% | 62.16% | 45.81% |
| Qwen3-Nothinking-8B | 3.98% | 2.91% | 34.78% | 5.38% | 79.92% | 25.39% |
| Diff (relative) | ↑644.97% | ↑1126.80% | ↑65.27% | ↑718.59% | ↓22.22% | ↑80.38% |
| Qwen3-Thinking-1.7B | 7.96% | 6.74% | 36.27% | 11.37% | 68.98% | 26.26% |
| Qwen3-Nothinking-1.7B | 3.10% | 2.73% | 38.46% | 5.10% | 96.78% | 29.23% |
| Diff (relative) | ↑156.77% | ↑146.89% | ↓5.69% | ↑122.94% | ↓28.72% | ↓10.16% |
| R1-Distill-Qwen2.5-32B | 40.71% | 38.43% | 72.01% | 50.12% | 76.96% | 55.65% |
| Qwen2.5-32B | 17.70% | 30.05% | 40.05% | 29.73% | 77.96% | 39.10% |
| Diff (relative) | ↑130.00% | ↑27.89% | ↑79.80% | ↑68.58% | ↓1.28% | ↑42.32% |
| R1-Distill-Qwen2.5-7B | 8.85% | 6.92% | 17.19% | 9.87% | 40.15% | 16.60% |
| Qwen2.5-7B | 16.81% | 14.39% | 11.97% | 13.07% | 7.59% | 12.77% |
| Diff (relative) | ↓47.35% | ↓51.91% | ↑43.61% | ↓24.48% | ↑428.99% | ↑30.00% |
| R1-Distill-Qwen2.5-1.5B | 0.88% | 0.36% | 1.15% | 0.55% | 40.28% | 8.64% |
| Qwen2.5-1.5B | 0.88% | 0.73% | 0.93% | 0.82% | 17.37% | 4.15% |
| Diff (relative) | 0.00% | ↓50.68% | ↑23.66% | ↓32.93% | ↑131.89% | ↑108.49% |
| R1-Distill-Llama-70B | 48.23% | 48.63% | 68.46% | 56.87% | 79.41% | 60.32% |
| Llama-70B | 29.20% | 55.37% | 28.73% | 37.83% | 19.18% | 34.06% |
| Diff (relative) | ↑65.17% | ↓12.17% | ↑138.29% | ↑50.33% | ↑314.03% | ↑77.09% |
| R1-Distill-Llama-8B | 20.35% | 19.85% | 42.75% | 27.11% | 41.31% | 30.27% |
| Llama-8B | 23.89% | 21.86% | 17.67% | 19.54% | 0.00% | 16.59% |
| Diff (relative) | ↓14.82% | ↓9.19% | ↑141.94% | ↑38.74% | ↑inf | ↑82.46% |

observed performance superiority of reasoning models in race detection is primarily introduced by qualitative improvements in their underlying reasoning logic and the ability to self-refine.

---

**Listing 2: Model response from DeepSeek-R1**

(...) Now, for x:
- The main thread writes x in line 25 (inside the loop) without any synchronization.
- The created thread reads x at line 15 (assert(x ==42)), which is inside the mutex.
So, the writes in line 25 are not protected, and the read in line 15 is protected. Since the main thread's writes are not synchronized, there's a data race between line 25 (write) and line 15 (read).

**But wait**, the main thread's loop writes x from 0 to 42. The created thread only reads x after x_set is 1. But when does x_set get set? The main thread sets x_set to 1 after the loop. So, the created thread's read of x (line 15) happens after x_set is 1. (...)

But when does the created thread read x? Only after x_set is 1. And x_set is set to 1 after the loop. So, the writes to x in the loop are all before x_set is set to 1. Therefore, the created thread's read of x occurs after all the writes. **Therefore, there's no data race on x**.

---

## O    THE LIMITATION OF STATIC ANALYSIS

Dealge, as a static analyzer, fundamentally cannot fully capture the dynamic behavior of programs. This inherent limitation, shared among all static analysis tools, frequently leads to the reporting of false positives and false negatives. For instance, a program may use a variable to restrict the number of loop iterations, the value of which cannot be precisely determined through static analysis

alone, often due to runtime uncertainty or dependence on the dynamic workload. To proceed with the analysis, static analyzers typically resort to heuristic estimations of the number of iterations, rules that are predefined by human experts. When these heuristics result in over-estimation or under-estimation, they may generate false positives or false negatives. We observe such cases throughout our DRDBench dataset, and we confirm that all false positives reported by Dealge on DRDBench are attributable to the inaccurate heuristic estimation of loop iterations.

---

**Listing 3: Model response from DeepSeek-V3**

Let's analyze the program step by step to identify any potential data race bugs.

1. **Thread Creation and Joining**:
- The main thread creates a child thread at line 23.
- The child thread executes the thread function (lines 11-17).
- The main thread joins the child thread at line 28.

2. **Shared Variables**:
- x (line 6): an integer variable.
- x_set (line 7): a boolean flag indicating whether x has been set.
- lock (line 4): a mutex used to protect access to x and x_set.
- cond (line 5): a condition variable used to signal changes to x_set.

3. **Accesses to Shared Variables**:
- **Child Thread (thread function)**:
- Locks lock at line 12.
- Waits on cond (line 14) until x_set becomes true (line 13).
- Reads x at line 15 (protected by lock).
- Unlocks lock at line 16.
- **Main Thread**:
- Writes to x in a loop at line 25 (unprotected by lock).
- Writes to x_set at line 26 (unprotected by lock).
- Signals cond at line 27 (unprotected by lock).

4. **Potential Data Races**:
- The main thread writes to x at line 25 without holding the mutex lock. The child thread reads x at line 15 while holding the mutex. These accesses can happen concurrently, leading to a data race.
- The main thread writes to x_set at line 26 without holding the mutex lock. The child thread reads x_set at line 13 while holding the mutex. These accesses can also happen concurrently, leading to a data race.

5. **Synchronization**:
- The mutex lock is used correctly in the child thread to protect the accesses to x and x_set.
- However, the main thread does not use the mutex to protect its writes to x and x_set, which is incorrect.

6. **Conclusion**:
- There are two data races in the program:
- Between the write to x at line 25 and the read at line 15.
- Between the write to x_set at line 26 and the read at line 13.

---

## P  AGGREGATION STRATEGY

This section first explains why the optimal aggregation strategy varies by model, and then shares discussions about our points of view in finding a new aggregation strategy to consistently achieve optimal model performance in future studies.

In traditional two-class classification tasks, models typically output probability scores for both positive and negative classes. We denote the probability score for the positive class as $p$ and for the negative class as $1 - p$.

In practical applications, the model needs to provide a specific class label instead of just a probability score. To achieve this, we can set a threshold, denoted as $t$. If the probability score $p$ is greater than or equal to the threshold $t$, we classify the input as belonging to the positive class. Conversely, if $p$ is below the threshold, we classify the input as the negative class. The threshold $t$ can be adjusted to balance the trade-off between precision and recall. The optimal threshold may vary depending on the model, and finding this optimal threshold typically requires extensive evaluation.

LLMs output probability scores for individual tokens. However, a data race, represented as a JSON object, consists of multiple tokens. Besides, the order of data races in the LLM's responses can vary. These make it challenging to directly calculate the probability of the LLM reporting a specific data race. One possible solution is to sample multiple responses from the LLMs and count how many times a specific data race appears in those responses. This allows us to estimate the probability $p$ that the LLM predicts a specific data race. We can then apply a threshold $t$ to determine whether the model should ultimately report a specific data race.

The different aggregation strategies proposed in our submission can be understood as various settings for the threshold $t$. Specifically, the strategy `Maj@k` can be seen as setting the threshold $t$ to $\frac{\lceil k \rceil}{2k}$, `Int@5` sets the threshold $t$ to 1.0, and `Uni@5` sets $t$ to $\frac{1}{k}$. When a large number of samples is taken, meaning $k$ is large, the threshold $t$ approaches 0.5 for `Maj@k`, 1.0 for `Int@5`, and 0.0 for `Uni@5`.

In traditional two-class classification tasks, the optimal threshold $t$ represents the best trade-off between precision and recall. It can differ from one model to another. In the context of data race detection, precision reflects the model's ability to correctly identify data races, minimizing false positives, while recall indicates the model's effectiveness in finding all actual data races, minimizing false negatives. Similar to the traditional two-class classification tasks, in our data race detection task, the optimal aggregation strategy may also differ based on the model, as different models prioritize precision and recall differently.

In this study, we follow the self-consistency approaches (Wang et al., 2023; Chen et al., 2024; Wu et al., 2025) to examine the effectiveness of three different voting strategies for aggregating responses. The evaluation results indicate that simple voting strategies do not consistently yield optimal performance, as the best aggregation strategy varies by model. This motivates us to explore how we can develop a new aggregation strategy that achieves optimal performance across various models.

From our perspective, we may begin by sampling multiple responses from a model. Next, we analyze the token-level probabilities of all tokens that belong to a specific data race to assess the model's overall "confidence" regarding each identified data race. Afterward, we combine the data race reports from all the sampled responses and use the model's "confidence" to filter out lower-quality reports. Since this study extends beyond the scope of this paper, we plan to investigate it in future work.

## Q    COMPARISON WITH CODING AGENTS

To rigorously evaluate the effectiveness of tool-calling agents in the race detection task, we conducted comparative experiments. We benchmarked LLMs both with and without the OpenHands scaffold. To eliminate the potential bias that may be introduced by our preprocessing steps, this comparison was performed on both the preprocessed and unpreprocessed versions of the programs. Due to the reason that only DeepSeek-V3 and the Qwen3 series models in our evaluated open-source LLMs supported the function calling capability required by the OpenHands scaffold, we only benchmarked two LLMs, DeepSeek-V3 and Qwen3-Thinking-32B (the best-performing Qwen3 series model in our main evaluation).

The evaluation results are presented in Tables 15 and 16. The effectiveness of the agent framework is observed to vary significantly by the underlying backbone model. Specifically, the Qwen3-Thinking-32B model suffers performance degradation when utilizing OpenHands, whereas DeepSeek-V3 demonstrates a slight but measurable improvement. Analysis of the interaction history reveals the root cause. Qwen3-Thinking-32B reduces both its thinking frequency and length when operating

Table 15: Evaluation of OpenHands on preprocessed programs.

| Model | Pass@1 | Recall | Precision | F1 | 1-FPR | Average |
|---|---|---|---|---|---|---|
| Qwen3-Thinking-32B | 36.36% | 39.73% | 65.91% | 49.57% | 80.77% | 54.47% |
| + OpenHands | 31.82% | 36.99% | 57.45% | 45.00% | 83.33% | 50.92% |
| Diff (relative) | ↓12.49% | ↓6.90% | ↓12.84% | ↓9.22% | ↑3.17% | ↓6.52% |
| DeepSeek-V3-671B | 36.36% | 60.27% | 38.94% | 47.31% | 50.00% | 46.58% |
| + OpenHands | 45.45% | 73.97% | 52.43% | 61.36% | 62.82% | 59.21% |
| Diff (relative) | ↑25.00% | ↑22.73% | ↑34.64% | ↑29.70% | ↑25.64% | ↑27.12% |

Table 16: Evaluation of OpenHands on original programs.

| Model | Pass@1 | Recall | Precision | F1 | 1-FPR | Average |
|---|---|---|---|---|---|---|
| Qwen3-Thinking-32B | 31.82% | 30.14% | 52.38% | 38.26% | 84.62% | 47.44% |
| + OpenHands | 18.18% | 24.66% | 45.00% | 31.86% | 83.33% | 40.61% |
| Diff (relative) | ↓42.87% | ↓18.18% | ↓14.09% | ↓16.73% | ↓1.52% | ↓14.41% |
| DeepSeek-V3-671B | 31.82% | 47.94% | 50.00% | 48.95% | 75.64% | 50.87% |
| + OpenHands | 45.45% | 67.61% | 61.54% | 64.43% | 70.51% | 61.91% |
| Diff (relative) | ↑42.83% | ↑41.03% | ↑23.08% | ↑31.62% | ↓6.78% | ↑21.70% |

within the OpenHands framework (illustrated as Listing 4), inhibiting its performance. Conversely, DeepSeek-V3 leverages task planning and self-reflection to enhance its race detection capabilities in multi-turn conversations (illustrated as Listing 5). Critically, both LLMs demonstrate no utilization of dynamic code utilities (such as the `gcc` compiler or `gdb` debugger) when working with the OpenHands scaffold, suggesting they still rely solely on static analysis even when dynamic tools are available. This absence of tool utilization strongly indicates a lack of domain-specific knowledge for appropriately associating the race detection task with these usable dynamic utilities.

Listing 4: Agent trajectory of Qwen3-Thinking-32B

```
USER_ACTION
// The task prompt, omitted
ACTION
// The commands for reading the code file(s), no reasoning content before taking the action
OBSERVATION
// The content of the code file(s), omitted
ACTION
// The commands for writing the results, together with the content to be written, no reasoning
content before taking the action
OBSERVATION
// The file is created
FINISH
```

Listing 5: Agent trajectory of DeepSeek-V3

**USER_ACTION**
// The task prompt, omitted
**ACTION**
// The commands for reading the code file(s), no reasoning content before taking the action
**OBSERVATION**
// The content of the code file(s), omitted
**ACTION**
AgentThink: Now I need to analyze this program for data race bugs. Let me carefully examine the shared variables and synchronization mechanisms. Let me analyze the program step by step: ... (details omitted)
**OBSERVATION**
Your thought has been logged.
**ACTION**
AgentThink: Let me examine the functions more carefully to see which ones use synchronization: ... (details omitted)
**OBSERVATION**
Your thought has been logged.
**ACTION**
AgentThink: Let me create a detailed analysis of all shared variable accesses and their protection: ... (details omitted)
**OBSERVATION**
Your thought has been logged.
**ACTION**
AgentThink: Now let me identify the potential data races by checking which unprotected accesses can happen concurrently: ... (details omitted)
**OBSERVATION**
Your thought has been logged.
**ACTION**
AgentThink: Let me also check if there are any races involving scull_quantum and scull_qset: ... (details omitted)
**ACTION**
// The commands for writing the results, together with the content to be written
**OBSERVATION**
// The file is created
**ACTION**
AgentThink: Let me verify the JSON file was created correctly:
// The commands for verifying the JSON structure, details omitted
**OBSERVATION**
// The Python interpreter says that the JSON file is valid
**FINISH**

Listing 6: Model response when the program contains only two *seed* instances

(...) But the real shared variable here is seed. Let's look at where seed is accessed. (...) Wait, the code for PseudoRandomUsingAtomic_nextInt has:
Line 39: assert(seed !=0);
Line 40: atomic_acquire();
Line 41: read = 1;
Line 42: nexts = calculateNext(read);
Line 43: seed = nexts;
Line 44: atomic_release();
Wait, the code here seems to set read to 1, then compute nexts based on read (which is 1), then assign seed to nexts. **But seed is only being written here. The read from seed is line 39**, which is outside the atomic_acquire. (...) **So two threads could read seed in line 39 while another is writing it in line 43.** (...)

```
01  #include <assert.h>
02  #include <pthread.h>
03  int m = 0;
04  pthread_mutex_t mutex = PTHREAD_MUTEX_INITIALIZER;
05  void atomic_acquire()
06  {
07      pthread_mutex_lock(&mutex);
08      while (m != 0)
09      {
10          pthread_mutex_unlock(&mutex);
11          pthread_mutex_lock(&mutex);
12      }
13      m = 1;
14      pthread_mutex_unlock(&mutex);
15  }
16  void atomic_release()
17  {
18      pthread_mutex_lock(&mutex);
19      while (m != 1)
20      {
21          pthread_mutex_unlock(&mutex);
22          pthread_mutex_lock(&mutex);
23      }
24      m = 0;
25      pthread_mutex_unlock(&mutex);
26  }
27  inline int calculateNext(int s2)
28  {
29      int cnex;
30      do
31          cnex = rand();
32      while (cnex == s2 || cnex == 0);
33      return cnex;
34  }
35  int seed = 1;
36  inline int PseudoRandomUsingAtomic_nextInt()
37  {
38      int read, nexts;
39      assert(seed != 0);
40      atomic_acquire();
41      read = 1; // `read = seed;` leads to failure
42      nexts = calculateNext(read);
43      seed = nexts;
44      atomic_release();
45      return 0;
46  }
47  void *thr1(void *arg)
48  {
49      PseudoRandomUsingAtomic_nextInt();
50      return 0;
51  }
52  int main()
53  {
54      pthread_t t;
55      while (1)
56      {
57          pthread_create(&t, 0, thr1, 0);
58      }
59  }
```

Figure 8: Changing a single code line can cause the rate of correctly detecting the data race between lines 39 and 43 to drop from 98% to 32% (DeepSeek-R1-671B) or from 40% to 11% (DeepSeek-V3-671B) under 100 samples.

## R  FAILURE MODES

Figures 8 and 9 present the full versions of the programs used to demonstrate the failure modes discussed in Section 6. In the following, we provide several deeper insights into these two failure modes.

In the failure mode illustrated in Figure 8, we find that the primary cause of the issue is the model's lack of awareness regarding the memory access of the variable *seed* at line 39 when the program contains more than two instances of the variable *seed*. Listings 6 and 7 show how the model responds when presented with programs that have two and three instances of *seed*, respectively. In the first example with two instances, the model correctly recognizes both accesses to the *seed* variable. When

```
01  #include <assert.h>
02  #include <pthread.h>
03  int flag1 = 0, flag2 = 0; // pthread_mutex_t flag1, flag2;
04  int x;
05  pthread_mutex_t mutex = PTHREAD_MUTEX_INITIALIZER;
06  void *thr1(void *_)
07  {
08      pthread_mutex_lock(&mutex);        // replace with the following
09      flag1 = 1;                         //   will be fine:
10      while (flag2 == 1)                 //
11      {                                  // pthread_mutex_lock(&flag1);
12          pthread_mutex_unlock(&mutex);  // pthread_mutex_lock(&flag2);
13          pthread_mutex_lock(&mutex);    //
14      }                                  //
15      pthread_mutex_unlock(&mutex);      //
16      x = 0;
17      return 0;
18  }
19  void *thr2(void *_)
20  {
21      pthread_mutex_lock(&mutex);        // replace with the following
22      flag2 = 1;                         //   will be fine:
23      while (flag1 == 1)                 //
24      {                                  // pthread_mutex_lock(&flag2);
25          pthread_mutex_unlock(&mutex);  // pthread_mutex_lock(&flag1);
26          pthread_mutex_lock(&mutex);    //
27      }                                  //
28      pthread_mutex_unlock(&mutex);      //
29      x = 1;
30      return 0;
31  }
32  int main()
33  {
34      pthread_t t1, t2;
35      pthread_create(&t1, 0, thr1, 0);
36      pthread_create(&t2, 0, thr2, 0);
37      pthread_join(t1, 0);
38      pthread_join(t2, 0);
39      return 0;
40  }
```

Figure 9: Replacing standard mutex APIs with user-defined synchronization causes the rate of false-positive data race reports (on variable $x$) to significantly increase from 0% to 92% (DeepSeek-R1) or 2% to 100% (DeepSeek-V3) under 100 samples.

a third instance is added, the model seems to concentrate on only two of the multiple instances. It still ignores the *seed* instance at line 39, even after refining its thinking by reflection (shown as the bold sentences in Listing 6). This ultimately leads to the detection failure. We guess that this issue arises because the Transformer's token-to-token attention mechanism struggles to simultaneously capture the relationships between multiple pairs of tokens, particularly when these token pairs have similar meanings or intentions.

---

Listing 7: Model response when the program contains three *seed* instances

(...) Therefore, the accesses to seed in lines 41 (read) and 43 (write) are protected by this custom lock. So there's no data race here. (...) **But what about other accesses to seed? Are there any other lines where seed is accessed?** Let's check the code. The function calculateNext (line 32) checks if cnex is equal to s2 (which is the seed passed in), but that's a local variable. **The seed is read in line 41 and written in line 43. So all accesses to seed are within the critical section.** (...)

---

In the program illustrated in Figure 9, the user utilizes two variables, *flag1* and *flag2*, along with a standard lock variable, *mutex*, to ensure exclusive access between the two functions, *thr1* and *thr2*. The models attempt to analyze the semantics of this code, which is presented in Listing 8. However, they only identify that the variable $x$ is not protected by the lock variable *mutex*, failing to realize that the values of *flag1* and *flag2* provide exclusive access. As a result, the models incorrectly report a data race on the variable $x$. When the user-defined synchronization is replaced with standard library APIs, the models can correctly recognize the exclusive access and do not report a data race. This situation highlights that the LLMs may not fully understand the code, which limits their capability to detect data races.

> **Listing 8: Model response when the program contains user-defined synchronization**
>
> (...) Starting with flag1. Thr1 writes to flag1 at line 9 while holding the mutex. Thr2 reads flag1 at line 23. But wait, when thr2 is in the loop (line 23), it's inside the mutex lock. Because thr2 is holding the mutex when it checks flag1, and thr1 also locks the mutex when modifying flag1. So any access to flag1 (both read and write) is protected by the mutex. So no data race on flag1.
> Similarly, flag2 is written by thr2 (line 22) under the mutex, and thr1 reads it at line 10. Again, both accesses are within the mutex, so no race there.
> Now, the variable x. Thr1 writes to x at line 16, and thr2 writes to x at line 29. Are these writes protected by the mutex? Let's check.
> In thr1, after exiting the loop, the thread unlocks the mutex (line 15) and then writes to x (line 16). Similarly, thr2 unlocks the mutex at line 28 and then writes to x at line 29. **So both writes to x happen after the mutex is unlocked. Therefore, the accesses to x are not protected by the mutex.** (...)

## S  LIMITATION

While the programs in the SV-Benchmarks cover a broad range of real-world applications and data race patterns, they cannot encompass every possible application and data race scenario. We aim to continuously expand the benchmark with new data races collected over time. Given that identifying and documenting data races is a time-consuming and labor-intensive process, our study primarily leverages existing datasets. Future work could extend the benchmark to include more recent data race vulnerabilities documented in sources such as the CVE[6] and CWE[7] databases. Besides, DRDBench contains only isolated concurrent programs, which may limit the generalizability of our findings to real-world software systems that often involve complex interactions between multiple components. Additionally, there is potential for exploring agent-based race detection approaches and repo-level data race detection techniques.

## T  BROADER IMPACTS

We hope this work sets a new baseline for evaluating LLMs' capabilities in data race detection. It has the potential to inspire future research on leveraging NNs and LLMs for effective and efficient data race detection, thereby improving the software quality of concurrent programs. At this moment, we do not foresee any obvious undesirable ethical or social impacts.

## U  LICENSE

The DRDBench is licensed under the Creative Commons Attribution 4.0 International License (CC BY 4.0), whose content is summarized below.

We release the benchmark under the CC-BY license and Terms of Use, requiring disclosure when used for model evaluation. This license supplements, but does not replace, the original licenses of source materials; compliance with these and any applicable rights of data subjects is necessary. This statement clarifies the responsibilities and liabilities associated with using this benchmark. While we've made every effort to ensure the samples' accuracy and legality, we cannot guarantee their absolute completeness or correctness. We assume no liability for any rights violations, whether legal or otherwise, that may occur through the use of this benchmark, including but not limited to copyright infringement, privacy violations, or misuse of sensitive information. By accessing, downloading, or using this benchmark, you implicitly accept this statement and agree to adhere to the terms and conditions of the CC-BY license. If you do not agree with these terms or the CC-BY license, you are not authorized to use this benchmark.

---

[6]https://www.cve.org/
[7]https://cwe.mitre.org/

## V   THE USE OF LLMS

This section clarifies the use of LLMs in our research study. In the dataset construction, we utilized the LLMs for helping to help annotate the data races (detailed in Section 3). In the evaluation experiments, we conducted experiments on open-source LLMs to assess their performance (detailed in Section 5). Additionally, we applied LLMs to polish the writing of this research paper.

