# OpenReview forum: "A Comprehensive Fine-Grained Evaluation of LLMs in Data Race Detection"
_ICLR.cc/2026/Conference — ICLR 2026 Conference Withdrawn Submission_

### Official Review · Reviewer_Nh9b · 2025-10-18

**Soundness:** 3
**Presentation:** 3
**Contribution:** 3
**Rating:** 6
**Confidence:** 3

**Summary:**

The authors propose DRDBench, a new pthreads-based dataset containing 1,003 C programs with precise location annotations for data race detection. Besides, they propose FineEval-Race a fine-grained evaluation framework that maps LLM outputs to individual ground-truth data races and measures LLMs' performance.

**Strengths:**

1.The argument that existing evaluations are too coarse is convincing.

2.The dataset is large, real-world (SV-Benchmark origins), and annotated to line granularity.

3.25 open-source LLMs (reasoning and non-reasoning variants) are evaluated, plus Deagle as a static analyzer baseline

4.The authors manually inspect common failure cases and identify interpretable failure modes.

**Weaknesses:**

1.The benchmark appears to be heavily pre-processed, potentially reducing its representativeness for real-world concurrent software.
(a) Removing comments may distort the natural structure of source code, as comments are integral to understanding program intent. While such preprocessing might be acceptable for controlled method evaluation, it undermines the realism of a benchmark.
(b) The evaluation setup, as suggested by the template prompts in Appendix E, seems to test LLMs only on isolated concurrent programs. However, real-world data races often emerge from inter-thread or inter-module interactions. The current benchmark design does not seem to assess the LLMs’ ability to handle such complex scenarios.

2.DRDBench includes only data race programs, which may limit the benchmark’s generalizability.

3.The benchmark lacks a systematic and labeled taxonomy of data race patterns. Without explicit categorization (e.g., read–write, write–write, synchronization-related races), it is difficult to assess whether the benchmark sufficiently covers the diversity of real-world race conditions or to analyze model sensitivity across pattern types.

4.The manuscript does not provide explicit statistics on thread counts or the distribution of concurrency complexity.

**Questions:**

See above.

---

> ### Author Response · Authors · 2025-11-18
>
> We sincerely thank the reviewer Nh9b for the supportive suggestions. Below are our responses to your concerns.
>
>
>
> ## Preprocessing
> The target of our research study is not limited to evaluating LLMs' performance on the specific data race detection task, but also includes investigating the general capabilities of LLMs in understanding concurrent programming logics and reasoning about concurrent program behaviors.
> Data race detection is conventionally regarded as a challenging task in software engineering, due to the complex interleavings of concurrent program executions.
> To correctly identify data races, one needs to (1) understand the synchronization concepts and mechanisms, (2) reason about the temporal causal order between concurrent operations, and (3) identify thread interleavings that can lead to data races.
> We believe this task can serve as a representative interface to evaluate the general capabilities of LLMs in understanding and reasoning about concurrent programs. **We have revised the abstract and Sections 1-2 to make this clearer.**
>
>
> We preprocessed the code to ensure that the models concentrate on understanding and reasoning about the concurrency logics of the programs, rather than being distracted by other factors, including file structure and code comments.
> We fully agree with the reviewer that it is important to evaluate the LLMs' performance in more realistic settings.
> We add three ablation studies to assess the impact of three individual preprocessing methods, and an additional experiment to evaluate LLM's performance on unpreprocessed code to evaluate their performance in real-world scenarios. **Please refer to Appendices D-G in the revised manuscript for details.**
>
> We agree with the reviewer that DRDBench contains only isolated concurrent programs, which may limit the generalizability of our findings to real-world software systems that often involve complex interactions between multiple components.
> **We have added this to the limitation section (Appendix S) of our paper revision.**
>
> ## DRDBench includes only data race programs
> We cannot fully understand this concern of the reviewer.
> Since our study specifically targets data race detection, would the reviewer be pleased to share insights on why including only data race programs in DRDBench will limit its generalizability?
> Besides, we believe that our findings on the LLMs' capabilities in understanding and reasoning about concurrent programs can provide generalizable insights.
>
>
> ## Race patterns
> We agree with the reviewer that providing statistics about the pattern categorization may provide useful insights.
> We use read-write and write-write patterns to categorize the data races.
> **The statistics are presented in Appendix C and Table 6.**
>
> ## Statistics on concurrency complexity
> As far as we know, there is no widely accepted metric to quantify the concurrency complexity of a concurrent program.
> We collect two types of statistics to share some insights about the concurrency complexity of programs in DRDBench.
> They are the number of threads and the number of shared variables.
> **We present the statistics in Appendix C and Table 6.**

---

> > ### Comment · Reviewer_Nh9b · 2025-11-19
> > **Response to Authors**
> >
> > Thank you for addressing my concerns and including additional experiments!
> >
> > I am inclined towards accepting this paper and I have updated the score to indicate the same

---

> > > ### Author Response · Authors · 2025-11-19
> > > **Thank you note**
> > >
> > > Thank you for your valuable suggestions and your generous support of our research!

---

### Official Review · Reviewer_893y · 2025-10-31

**Soundness:** 1
**Presentation:** 2
**Contribution:** 2
**Rating:** 2
**Confidence:** 2

**Summary:**

This paper proposes a novel benchmark dataset comprising 1003 p-threads programs derived from SV-COMP, along with a new evaluation framework, FineEval-Race, designed to assess the performance of data race detection in LLMs. The authors conduct a comparative study of 25 different LLMs based on their proposed dataset and evaluation framework.

**Strengths:**

The paper’s comparative study shows a wide range of comparisons between each model, and empirically shows that the reasoning models perform better than non-reasoning models for the data race detection task. Such a comparison has significant meaning to the software engineering community and AI community when it comes to choosing a base model for downstream tasks that require reasoning about the correctness of pthreads programs. The evaluation framework proposes to use metrics based on well-established metrics such as precision, F1, and recall, which many community members in the software engineering would agree upon.

**Weaknesses:**

Although the evaluation framework metric uses aggregated results of well-established metrics, there is a substantial shortcoming of the ranking metric, the S score. The authors define the S score of a large language model D as “the sum of its rankings across all metrics”, where part of the metric consists of recall, precision, F1, and False Positive Rates. However, precision, F1, and recall all rely on the number of detected true positives, so there are three terms that utilize true positives, but there is only one term that depends on the number of false positive detections. Therefore, an unweighted sum of the rankings based on the proposed metrics is geared towards favoring the models with high true positive detection. This could lead to unfair comparison of two models where they both achieve similar performance with respect to true positive detections, but have a significant difference in their ability to avoid false positives. For example, in the Greedy decoding of Table 1, DeepSeek-V3-671B and Qwen2.5-72B have similar performance in terms of precision, but their performance drastically differs for FPR, yet both of their S score are 33. Hence, it is insufficient to compare the models with the naive summation of the rankings from each metric.

**Questions:**

One suggestion for designing a fair metric is to compare the importance of each metric (F1, precision, FPR, etc.) and construct the S score as the weighted sum of the relative importance. Please refer to the Weaknesses section for questions.

---

> ### Author Response · Authors · 2025-11-18
>
> We sincerely appreciate Reviewer 893y for the constructive feedback. We respectfully argue that the concerns raised regarding our evaluation methodology and scoring system appear to stem from a misinterpretation of our evaluation metrics and a reliance on potentially inaccurate data points in the provided example. Our detailed responses follow below.
>
> ## Clarification Regarding the Precision and FPR Metrics
>
> The reviewer expresses concern that the S score leads to an unfair comparison because "there are three terms that utilize true positives, but there is only one term that depends on the number of false positive detections." We respectfully clarify that this may stem from a misunderstanding of how the precision and False Positive Rate (FPR) metrics are defined and partitioned within our evaluation framework.
>
> We designed our framework to distinguish between two categories of false positives ($FP$):
> 1. $FPa$: False positives reported on programs that contain data races (raceful programs).
> 2. $FPb$: False positives reported on programs that are free of data races (race-free programs).
>
> Since true positive ($TP$) data races can only occur on raceful programs, we design the recall and precision metrics to be calculated solely on those programs.
> That is, the precision metric is calculated by $\frac{TP}{TP+FPa}$, which is consistent with the definition provided in Section 4. This metric is dependent on both $TP$ and $FPa$.
> We believe the reviewer may have assumed the precision metric included $FPb$, calculating it as $\frac{TP}{TP+FPa+FPb}$.
> Additionally, the FPR metric was introduced specifically to evaluate the LLMs' hallucination on race-free programs.
> This metric is, therefore, dependent solely on $FPb$, as demonstrated by its definition in Section 4. We believe the reviewer may have assumed the FPR metric includes both $FPa$ and $FPb$.
>
> The recall metric is dependent solely on $TP$.
> Besides, the F1 score is the harmonic mean of precision and recall; it is dependent on $TP$ and $FPa$.
> Given the definitions of precision and FPR discussed above, we argue that the four metrics (recall, precision, F1, and FPR) are fairly balanced and independently measure distinct aspects of performance. The reviewer's criticism that "an unweighted sum of the rankings based on the proposed metrics is geared towards favoring the models with high true positive detection" may not fully capture the balanced nature of our defined metrics.
>
> ## Correction of Data Points and Scope in the Reviewer's Example
> The reviewer provided an insightful example to illustrate a perceived unfairness in the S score design. However, we must point out that this example contains several critical factual errors regarding our published evaluation results and the scope of the S score definition, which significantly affect its conclusion.
>
> 1. The reviewer's example states that the S scores for DeepSeek-V3-671B and Qwen2.5-72B are both 33. We must respectfully clarify that, according to the evaluation results presented in our Table 1, their actual S scores are 142 and 161, respectively. This significant difference completely changes the premise. It may invalidate the reviewer's conclusion drawn from the example.
>
> 2. The reviewer seems to have calculated the S score based on the sum of rankings from only four evaluation scores (recall, precision, F1, and FPR in the greedy decoding setting). We respectfully point out that this is an incomplete view of the S score definition. The penultimate paragraph in Section 4 clearly states: "This results in a total of 18 unique evaluation scores. (...) The S score for a model $D$ is then calculated as the sum of its rankings across all metrics." Thus, the S score is calculated by summing up the rankings from all 18 evaluation scores, instead of the 4 scores denoted by the reviewer.
>
> 3. Even applying the reviewer's simplified understanding of the S score (sum of rankings from four metrics), the conclusion that identical S scores are unreasonable overlooks the necessary balance of performance across metrics. The reviewer posits that since Qwen2.5-72B significantly outperforms DeepSeek-V3-671B in FPR, its S score (lower is better) should be significantly lower, given similar precision performance. This argument ignores that DeepSeek-V3-671B outperforms Qwen2.5-72B for the other two metrics: recall and F1. Given that DeepSeek-V3-671B performs better in two metrics (recall and F1) while Qwen2.5-72B significantly outperforms DeepSeek-V3-671B in only one metric (FPR), it remains justifiable for the two models to have comparable, or at least not drastically different, final S scores (rankings), as the overall ranking balances these tradeoffs.

---

> ### Author Response · Authors · 2025-11-27
> **Looking forward to your feedback!**
>
> Dear Reviewer 893y,
>
> Thank you again for your thoughtful review and for the time you've dedicated to our work. We have added clarifications to resolve your concerns. As the discussion period comes to a close, we wanted to follow up and see if our responses have fully addressed your concerns. We are looking forward to hearing from you.
>
> Thank you again for your attention.
>
> Best regards,
>
> Authors

---

### Official Review · Reviewer_7ukv · 2025-11-01

**Soundness:** 2
**Presentation:** 3
**Contribution:** 3
**Rating:** 4
**Confidence:** 5

**Summary:**

The paper releases DRDBench, a benchmark of 1,003 pthread-based C programs containing data race programs or race-free programs and an evaluation framework FineEval-Race to evaluate data race detection with LLMs. The construction pipeline includes user-defined header inlining, comment removal, formatting for one-statement-per-line, human+tool annotation, and meta-review. Experiments span strong reasoning and non-reasoning LLMs with pass@k, F1, and FPR. The authors also analyze common failure modes (e.g., misunderstanding user-defined synchronization).

**Strengths:**

1. Clear pipeline for constructing data race benchmarks with fine-grained line annotations and a meta-review process.

2. FineEval-Race task design provides a concrete, reusable framework for model evaluation on data race detection.

3. Comprehensive evaluation of open-sourced LLMs on the detection accuracy as well as the hallucination rate.

**Weaknesses:**

1. Key preprocessing choices (comment removal, single-file flattening) are insufficiently justified relative to their impact on LLM behavior.

2. No agentic/tool-use baselines despite a task that naturally benefits from tools.

3. Findings from the evaluation are mostly descriptive; audience and actionable guidance are under-specified.

**Questions:**

Thank you for the submission. I believe this is a timely benchmark that evaluates LLMs on the task of data race detection. The writing is easy to follow. However, there are a few drawbacks that I think the paper should address before getting accepted.

I do not fully understand the rationale for removing comments, given comments can encode developer intent and explanation that LLMs leverage. The paper states comments “may introduce noise” and are therefore removed, but provides no good ablation study to justify this design choice. This omission matters because the prompt explicitly teaches concurrency rules and relies on natural-language guidelines elsewhere.

Second, flattening multi-file projects into a single file is also debatable. The appendix comparison suggests minimal aggregate impact, but in Table 6, I see examples where the pass rate could reduce by 100% or increase by 50%. Also, using a single file to evaluate seems to simplify the task as it can reduce the code length. However, in the related work, the drawbacks of prior benchmarks seem to be over-simplifying the detection task and having too simple programs. So I feel this design choice does not fit with the goal of the paper to construct a comprehensive data race detection benchmark.

There is also a lack of evaluation on the performance of agents with tool uses. Data race detection is a suitable task for tool-calling agents. This is because requiring only the static analysis tools is not sufficient (the paper also reports Deagle having worse performance than the best open-sourced models). I expect agentic solutions with tools such as a debugger and filesystem handlers to have better performance, as they provide dynamic debugging information to the agents.

The paper presents a comprehensive evaluation of open-sourced models' performance on the data race detection tasks, but the findings remain relatively straightforward. For example, Table 1 shows that larger models with reasoning capabilities generally perform better. Such findings are not so different than those from conventional coding benchmarks. There is a lack of actionable items or insights that the system could give.

Technical Questions:

1. Can you provide an ablation across programs for the LLM detection accuracy with and without comments removal? How do different comment types (docstrings, TODOs, etc) affect outcomes?

2. Is it possible to evaluate on cross-file repository setup that makes the data race detection task more realistic?

3. For the annotation process, why do we first let the human annotator review the outputs and then independently write annotations, but not vice versa? Would that lead to biases?

4. Does meta-reviewer review all of the annotations generated or only part of them?

5. The framework only checks if the line number matches with the ground-truth results, but isn't that going to lead to more false positives if there are multiple variables in a line?

6. Who is the target audience of the benchmark and the framework? What are some actionable insights we can get from the evaluation?

---

> ### Author Response · Authors · 2025-11-18
> **Rebuttal (1/2)**
>
> We sincerely thank the reviewer 7ukv for the professional comments and suggestions. Before answering the questions raised in your review, we would like to clarify the target of our research study, which is not limited to evaluating LLMs' performance on the specific data race detection task, but also includes investigating the general capabilities of LLMs in understanding concurrent programming logics and reasoning about concurrent program behaviors.
>
> ## The target of our research study
>
> Data race detection is conventionally regarded as a highly challenging task in software engineering, primarily due to the complex interleavings of concurrent program executions. To correctly identify data races, an analyzer must perform a sequence of intricate steps:
> - understanding the synchronization concepts and mechanisms,
> - reasoning about the temporal causal order between concurrent operations,
> - identifying thread interleavings that can lead to data races.
>
> Given these demands, we believe data race detection can serve as a representative interface to evaluate the general capabilities of LLMs in understanding and reasoning about concurrent programs. **We have revised the abstract and Sections 1-2 to make this clearer.**
>
> The FineEval-Race framework and the line-level annotation within DRDBench are specifically designed to rigorously evaluate the concurrency understanding and reasoning capabilities of LLMs. This design critically distinguishes DRDBench from existing benchmarks.
> Current benchmarks often simplify the race detection task into a program-level binary classification problem, a coarse granularity that is insufficient for comprehensively evaluating LLM capabilities in concurrent program analysis. This limitation arises because, under program-level classification, it is difficult to reliably distinguish whether model success is due to stochastic guessing or a genuine comprehension of the intricate details of concurrent program behaviors.
> Our fine-grained evaluation effectively mitigates this risk by requiring models to identify the exact lines of code involved in data races. Models that lack a true understanding and robust reasoning about concurrent program behaviors are highly likely to fail this granular requirement, resulting in the production of false positives or false negatives. Thanks to your suggestion, **we have added a clarification of this point in Section 2 of our revision**.
>
> Due to the reason that our ultimate target is to evaluate the general capabilities of LLMs in understanding and reasoning about concurrent programs, we introduced several simplifications in our evaluation.
> We chose to aggregate the code into a single file to ensure that the models concentrated on understanding the concurrency logic of the programs, rather than being distracted by the file structure.
> Besides, we removed the comments in the source code to ensure the models comprehended and reasoned the concurrency logic based on their own capability, rather than relying on the comments that may directly describe the concurrency logic.
> We also chose not to evaluate the tool-calling agents because our target was to evaluate the inherent capabilities of LLMs in understanding and reasoning about concurrent programs, rather than their capability of using tools.
>
>
> Thanks to the thoughtful comments from the reviewer, we agree that it is still important to evaluate the performance of LLMs in more realistic settings.
> **We have added ablation studies to all three preprocessing methods.**
> These ablation studies assess the impact of file flattening, code removal, and code formatting on LLM's performance.
> Besides, **we have added an additional evaluation on the unpreprocessed code to evaluate LLM's performance in real-world scenarios**.
>
>
> We agree that the capability of using the tool can also be a part of LLM's capability for understanding and reasoning about concurrent programs.
> Thus, **we conduct a comparative experiment between LLMs and agents to investigate whether the tool-using capability can help LLMs achieve better performance**. Due to the word limit, we detail them in the next note.

---

> ### Author Response · Authors · 2025-11-18
> **Rebuttal (2/2)**
>
> Below are our responses to the technical questions.
>
> ## Ablation studies
> We are pleased to add an experiment to assess the influence of removing code comments on the LLMs' performance.
> **Please refer to Appendix E in the revised manuscript for the detailed experimental setup and results.**
> The conclusion is, removing the code comments has a limited impact on the model's performance: the relative performance improvement or degradation typically does not exceed 10\%.
>
> In addition, **please refer to Appendix D for the ablation study on header inlining and  Appendix F for that on code formatting.** These studies confirm that preprocessing has a limited (<15%) impact on model performance.
>
> The programs in DRDBench do not contain specialized comments such as docstrings or TODOs.
> Only two types of comments are present in the code: (1) a heading comment block declaring the rights and authors, and (2) single-line comments that explain the purpose or cautions of nearby code blocks.
> In our ablation study, we only try to either retain or remove all the comments.
>
> ## Evaluation on unpreprocessed code and tool-calling agents
> We add experiments to assess the LLMs' performance on unpreprocessed code, where the code is organized in its original file structure, contains comments, and has no formatting.
> The conclusion is, preprocessing has a minimal impact on non-reasoning models, but it can significantly improve the performance of reasoning LLMs.
> However, it is essential to note that this observed bias does not overturn the core insights obtained in our main evaluation.
> **Please refer to Appendix G in the revised manuscript for the detailed experimental setup and results.**
>
> We also add two experiments to evaluate the performance of tool-calling agents on both unpreprocessed code and preprocessed code.
> We obtain several actionable insights from the experiments.
> **Please refer to Section 5 and Appendix Q in the revised manuscript for the detailed experimental setup and results.**
>
> ## Human annotation under the assistance of tools
> Thanks for pointing this out.
> This is because data race detection is fundamentally challenging.
> Without reviewing the outputs of tools, human annotators usually need to spend hours on annotating a single program.
> To ensure the efficiency of human annotation, we allow the human annotators to refer to the outputs of tools to obtain an initial understanding of the code, especially the reasoning content generated by the two LLMs.
> To avoid the risk of being misled by tool outputs, we recruit annotators with strong backgrounds (more than 3 years of experience in concurrent programming).
> Besides, we specifically instruct them not to blindly trust the race annotations of the tools, but to carefully annotate based on their own understanding of the code.
> Furthermore, we conduct a meta-review process to help eliminate potential mistakes in the annotations.
> We believe these can guarantee unbiased annotation.
> **We have added these clarifications to Section 3 in the revision.**
>
>
> ## Meta review
> The meta reviewers always review all the annotations submitted by the primary human annotators and tools.
> They focus on rejecting annotations that are inconsistent with the actual concurrency logics.
> This verification process is simpler than the original annotation task, so the meta reviewers can efficiently review a large number of annotations.
> We sincerely thank the reviewer for the suggestion to clarify this point; **we have added additional clarification in Section 3 of our revision**.
>
>
> ## Line-level comparison and false positives
> The reviewer may be concerned that our evaluation framework may overestimate the precision of LLMs in the following scenario.
> Two variables (denoted as x and y) appear in the same two code lines, whereas only x is involved in a data race, e.g., the variable y is only read in both code lines.
> Under our evaluation framework, only the line numbers are checked against the ground truth.
> It may be possible that the LLM reports a false positive data race on y, whereas our framework incorrectly treats it as a true positive since the line number matches.
>
> We fully agree with the reviewer about the potential risk of our line-level evaluation framework for overestimating the precision of LLMs.
> However, we argue that our line-level evaluation is a trade-off between minimizing false positives and false negatives.
> Thanks to the suggestion from reviewer 7ukv, **we have revised the paper to include an additional discussion about this risk and the trade-off rationale behind our design choice.
> This can be found in Appendix J of the revised manuscript.**
>
> ## Target audience and actionable insights
> This research study aims to provide actionable insights for both software engineers and LLM developers.
> **We have added two explicit *actionable insights* paragraphs in Section 5 to summarize the key findings for both audiences.**
> We thank the reviewer for this suggestion, as it can help our study stand out.

---

> ### Author Response · Authors · 2025-11-27
> **Looking forward to your feedback!**
>
> Dear Reviewer 7ukv,
>
> Thank you again for your thoughtful review and for the time you've dedicated to our work. We have added discussions and experiments to resolve your concerns. As the discussion period comes to a close, we wanted to follow up and see if our responses have fully addressed your concerns. We are looking forward to hearing from you.
>
> Thank you again for your attention.
>
> Best regards,
>
> Authors

---

### Official Review · Reviewer_Ht39 · 2025-11-01

**Soundness:** 2
**Presentation:** 3
**Contribution:** 3
**Rating:** 6
**Confidence:** 3

**Summary:**

This work makes a contribution for evaluating large-scale language models in the complex task of data-race detection. The authors introduce a high-quality benchmark, DRDBench, and a fine-grained evaluation framework, FineEval-Race, shifting the assessment from coarse, program-level “yes/no” verdicts to precise measurement of individual data races with metrics such as precision and recall. This refined approach yields a deeper and more nuanced understanding of current model capabilities. Experiments on 25 open-source models, a comparison against the latest static analyzer, and insightful failure analyses ensure both rigor and impact. The findings—that reasoning-oriented models excel and that specific failure modes can be identified—provide clear and valuable guidance for future research.

**Strengths:**

1. The core idea of shifting evaluation from the program level to the individual data race level is the paper's greatest strength. This fine-grained approach is essential for a meaningful assessment of progress in this complex reasoning task and sets a new standard for future work.
2. DRDBench is built on solid ground, harvesting pthread cases from SV-Benchmarks to fill a critical gap, and its meticulous, multi-reviewer annotation process yields a ground truth we can trust.
3. The paper impressively evaluates 25 language models against the strong Deagle baseline. Its sharp analysis exposes a clear reasoning-vs-non-reasoning gap, models’ conservative–aggressive swings, and the promise–instability trade-off of sampling.

**Weaknesses:**

1. The programs in DRDBench, while sourced from real-world projects and verification suites, are relatively small in scale (max 624 Lines of Code). The performance of LLMs on larger, more complex, and sprawling codebases remains an open question.

**Questions:**

1. The results show Deagle achieving high recall (87.34%) but relatively low precision (63.82%), implying a notable number of false positives. Could you comment on the nature of these false positives from a state-of-the-art static analyzer on your benchmark? Does this suggest limitations in static analysis for certain concurrency patterns present in DRDBench?
2. The paper mentions using clang-format to ensure one statement per line. How significantly did this reformatting alter the program structure and line counts compared to the original source code? Is there a risk that this standardization inadvertently simplifies the task or changes it in a way that wouldn't reflect performance on unformatted, "in-the-wild" code?
3. Is the performance advantage of reasoning models driven primarily by longer output lengths, or by qualitative improvements in reasoning logic? A direct comparison of solution chains from top-performing reasoning versus non-reasoning models on identical complex cases would clarify this distinction.

---

> ### Author Response · Authors · 2025-11-18
>
> We sincerely thank the reviewer Ht39 for the thoughtful comments and suggestions. Below are our responses to the questions.
>
> ## **False positives of Deagle**
>
> Dealge, as a static analyzer, fundamentally cannot fully capture the dynamic behavior of programs.
> This inherent limitation, shared among all static analysis tools, frequently leads to the reporting of false positives and false negatives.
> For instance, a program may use a variable to restrict the number of loop iterations, the value of which cannot be precisely determined through static analysis alone, often due to runtime uncertainty or dependence on the dynamic workload.
> To proceed with the analysis, static analyzers typically resort to heuristic estimations of the number of iterations, rules that are predefined by human experts.
> When these heuristics result in over-estimation or under-estimation, they may generate false positives or false negatives.
> We observe such cases throughout our DRDBench dataset, and we confirm that all false positives reported by Dealge on DRDBench are attributable to the inaccurate heuristic estimation of loop iterations.
> **We have added this discussion as Appendix O in the revision.**
>
> ## **Impact of code formatting**
> Clang-format is a code formatting utility based on the Clang C/C++ compiler front-end.
> It can automatically adjust indentation, spacing, and line breaks according to a unified style, which was set to the `microsoft` style in our experiments.
> It is designed to ensure that the programming logic and execution semantics are preserved.
> The formatting results in a minor reduction in code length, with programs averaging approximately 160 lines of code (LOC) before formatting and approximately 140 LOC afterward.
> We further manually review the formatted code to ensure that the program's functionality and logic remain entirely equivalent to the original version. **We add this clarification to Section 3 in the revision.**
>
> We agree with the reviewer Ht39 that evaluating LLMs solely on formatted code presents a potential risk of not fully reflecting performance on unformatted code.
> To mitigate this, we conducted an additional ablation study to quantify the impact of code formatting on race detection performance.
> This experiment evaluated LLMs on both unformatted and formatted code.
> The results confirm that standardizing the coding styles typically has a limited impact on the model's race detection performance: the relative performance improvement or degradation does not exceed 15\%.
> This finding suggests that while formatting introduces a marginal bias, it does not fundamentally alter the core conclusions of our study.
> **Please refer to Appendix F in the revised manuscript for the detailed experimental setup and results.**
>
>
> ## **Comparison between reasoning and non-reasoning models**
>
> We agree with the reviewer Ht39 that investigating the root cause of the performance difference between reasoning and non-reasoning models can provide valuable insights.
> We conduct a case study to illustrate this.
> **Please refer to Appendix N in the revised manuscript for the detailed analysis.**

---

> ### Author Response · Authors · 2025-11-27
> **Looking forward to your feedback!**
>
> Dear Reviewer Ht39,
>
> Thank you again for your thoughtful review and for the time you've dedicated to our work. We have added discussions and experiments to resolve your concerns. As the discussion period comes to a close, we wanted to follow up and see if our responses have fully addressed your concerns. We are looking forward to hearing from you.
>
> Thank you again for your attention.
>
> Best regards,
>
> Authors

---

### Author Response · Authors · 2025-12-01
**General response (1/3)**

We sincerely appreciate the four reviewers, the ACs, the SACs, and the PCs for their time and effort in assessing our manuscript. We are grateful for the constructive feedback and insightful comments provided, which have significantly contributed to enhancing the quality of our work. As the rebuttal period is coming to a close, we would like to summarize our responses to the reviewers' concerns and present the key revisions made to the manuscript (highlighted in red in the PDF).
We hope this could help clarify any remaining questions and demonstrate our commitment to addressing the reviewers' concerns.

### **Strengths highlighted by reviewers**
- The core idea of shifting evaluation from the program level to the individual data race level is the paper's greatest strength. This fine-grained approach is essential for a meaningful assessment of model capabilities. This is highlighted by all four reviewers.
- DRDBench is built on solid ground, harvesting pthread cases from SV-Benchmarks to fill a critical gap. This is praised by Reviewers Ht39 and Nh9b.
- The extensive experiments across 25 open-source models provide a thorough evaluation of current capabilities. The comparison with state-of-the-art static analyzers adds further depth. This is noted by all four reviewers.
- FineEval-Race task design provides a concrete, reusable framework for model evaluation on data race detection. This is denoted by Reviewer 7ukv.
- The failure analysis offers valuable insights into common failure cases, identifying interpretable failure modes. This is appreciated by Reviewer Nh9b.

### **Concerns and our responses**
> **(Reviewers Ht39, 7ukv, and Nh9b) The rationale for introducing the three preprocessing steps**.

Reviewers Ht39, 7ukv, and Nh9b pointed out that the programs evaluated in our benchmark underwent several preprocessing steps, including flattening the file structure, removing comments, and formatting the code. They commented that they cannot find a clear rationale for these preprocessing steps in the benchmark construction pipeline. They also expressed concerns that these preprocessing steps might influence the model performance, and suggested conducting ablation studies to assess their impact.

We sincerely thank the reviewers for raising this important point and appreciate this opportunity to clarify the high-level target of our work.
We want to first clarify that the ultimate goal of our benchmark is not only limited to evaluating model performance on data race detection, but also to provide an assessment of model capabilities on understanding and reasoning of concurrent structures about concurrent programs.
The fine-grained evaluation framework we proposed focuses on assessing model capabilities for identifying individual data races.
It directly targets the goal of assessing model capabilities in reasoning about concurrent constructs.
This makes our benchmark and evaluation framework stand out from prior data race benchmarks that primarily focus on program-level binary verdicts, because the program-level binary verdicts may not provide insights into model capabilities on reasoning about concurrent constructs.
Besides,  our benchmark and evaluation framework offer a unique and necessary perspective for evaluating code execution comprehension of LLMs within the domain of concurrent programs, which fulfills an important gap that most existing code comprehension benchmarks focus solely on sequential programs.
In revision, we have added clarifications in **the abstract and Sections 1-2** to emphasize this high-level target of our work.

To align with this high-level target, we designed the three preprocessing steps to eliminate potential confounding factors that could obscure the assessment of model capabilities in reasoning about concurrent constructs.
The preprocessing steps have three purposes: (1) to eliminate risks of LLMs being distracted by file/code structures that are irrelevant to the target of assessing LLM's comprehension about concurrent constructs, and (2) to prevent LLMs from leveraging the external understanding of the concurrent constructs encoded in code comments to "cheat", and (3) to ensure the line number can serve as a reliable mechanism for locating data races.
We have revised **Sections 3** to add explanations for these rationales.


We appreciate the reviewers for pointing out that it is important to assess the impact of these preprocessing steps on model performance, especially for the purpose of assessing the capabilities of LLMs on detecting data races in real-world codebases.
We have conducted three ablation studies to evaluate the individual effects of each preprocessing step. Additionally, we performed another ablation study to assess the combined impact of all three preprocessing steps on model performance.
These four ablation studies have been added to the revised manuscript in **Appendices D-G**.

---

### Author Response · Authors · 2025-12-01
**General response (2/3)**

> **(Reviewer Ht39) Discussion on the limitation of the static analyzers**.

Reviewer Ht39 suggested that we need to discuss the limitations of static analyzers more thoroughly, especially given that our evaluation showed that LLMs can outperform them in certain scenarios.
We appreciate this insightful suggestion.
In revision, we have added a dedicated **Appendix O** to elaborate on the challenges and limitations of static analyzers on the data race detection task, with an analysis of why the static analyzers can have a high false positive rate on our DRDBench dataset.
We believe this addition provides a more balanced perspective on the comparative performance of LLMs and static analyzers.

> **(Reviewer Ht39) Comparison of solution chains from top-performing reasoning versus non-reasoning models**.

Reviewer Ht39 suggested comparing the solution chains generated by top-performing reasoning models against those from non-reasoning models to demonstrate the reason behind their performance differences.
We are pleased to conduct this comparison and have included two case examples with a qualitative analysis in **Appendix N** of the revised manuscript.
The case study demonstrates that the advantage of reasoning models indeed stems from their reasoning logic, which allows them to systematically analyze the code and identify data races.

> **(Reviewer 7ukv) Agentic baselines**.

Reviewer 7ukv suggested that we include agentic baselines in our evaluation to provide a more comprehensive comparison of model capabilities. We appreciate this feedback and have conducted experiments using agentic baselines with the OpenHands scaffold, as well as the two best-performing LLM backbones from our previous evaluations: Qwen3-Thinking-32B and DeepSeek-V3.
The details of the experimental setup and the results can be found in **Section 5 and Appendix Q** of the revised manuscript.

> **(Reviewer 7ukv) Details of the meta reviewing in the annotation process**.

Reviewer 7ukv raised several technical questions about the meta-reviewing process used during the annotation of data races. We have revised **Section 3** to provide more detailed explanations of the rationales for (1) allowing annotators to refer to tool outputs, and (2) letting the meta-reviewers review all the annotations submitted by human annotators and tools.


> **(Reviewer 7ukv) Determine whether the model answer is consistent with the ground truth based solely on the line number can lead to false positives**.

We appreciate Reviewer 7ukv for this insightful comment.
However, we would like to clarify that the line-level comparison choice was made for the trade-off between minimizing false positives and false negatives in our evaluation framework.
We have added **Appendix J** to detail the analysis of potential false positives and false negatives in our evaluation framework, along with the rationales for this design choice.

> **(Reviewer 7ukv) Target audiences and actionable insights**.

Reviewer 7ukv suggested that we clarify the target audiences of our benchmark and provide actionable insights for further research. We have revised **the abstract** to explicitly outline the intended audiences. Besides, we have detailed the actionable insights that are sourced from our evaluation results in **Section 5**.

> **(Reviewer 893y) Unfair metrics**.

Reviewer 893y raised concerns about the fairness of our evaluation metrics, suggesting that our current metrics are geared towards favoring the models with high true positive detection.
While we appreciate the reviewer for carefully examining our metrics, **we would like to clarify that the reviewer's concern stems from a misunderstanding of our evaluation metrics and factual errors regarding our evaluation results**.
For example, the reviewer claimed that "DeepSeek-V3-671B and Qwen2.5-72B have the same S score 33"; however, according to our evaluation results in **Table 1**, DeepSeek-V3-671B has an S score of 142, while Qwen2.5-72B has an S score of 161.
We have detailed the misunderstanding and the factual errors in our response to Reviewer 893y.

> **(Reviewer Nh9b) Data race patterns and thread counts of programs**.

Reviewer Nh9b suggested that we provide more detailed statistics on the data race patterns and the number of threads in the programs included in our benchmark.
We appreciate this suggestion and have added detailed statistics in **Appendix C and Table 6** of the revised manuscript.


### **Reviewer responses before the rebuttal cutoff**

Before the rebuttal cutoff, we received a response from **Reviewer Nh9b on** **19 Nov 2025**, which acknowledged our rebuttal and **raised the score from 6 to 8**.
The other three reviewers have not responded before the rebuttal cutoff.

---

### Author Response · Authors · 2025-12-01
**General response (3/3)**

In summary, before 27 Nov 2025, the paper's ratings were 2, 4, 6, 8, with confidence scores of 2, 5, 3, 3, respectively. It is essential to note that **the review comments from Reviewer 893y (rating 2, confidence 2) stemmed from a misunderstanding of our evaluation metrics and factual errors regarding our evaluation results**. We hope this critical information can be taken into account during the final decision-making process.

We thank all reviewers, ACs, SACs, and PCs again for their valuable feedback and consideration of our work. We believe the revisions made have significantly improved the manuscript, and we look forward to the opportunity to contribute to the ICLR community.

---

### Note · Authors · 2026-01-27

I have read and agree with the venue's withdrawal policy on behalf of myself and my co-authors.

---

### Meta-Review · Area_Chair_VjZB · 2026-01-09

**Summary:**

The paper introduces DRDBench, a benchmark for data race detection consisting of 1,003 programs, and FineEval-Race, a fine-grained evaluation framework. The authors evaluated 25 LLMs and compared them against static analyzers. While Reviewer Nh9b raised their score to 8 following the rebuttal , the overall reception remains mixed to negative, with other scores sitting at 2, 4, and 6. The primary concerns driving this decision revolve around the validity of the benchmark construction—specifically the heavy preprocessing steps like comment removal and file flattening—and the fairness of the proposed "S score" evaluation metric. Despite the authors' extensive rebuttal and additional ablation studies, the lack of engagement from the critical reviewers (7ukv and 893y) leaves these substantial concerns regarding the benchmark's realism and the metric's reliability unresolved, resulting in an average score that falls below the acceptance threshold for ICLR.

**Reviewer Concerns:**

The most critical outstanding concerns stem from Reviewer 7ukv and Reviewer Ht39 regarding the preprocessing pipeline used to construct DRDBench. Reviewers questioned the rationale and impact of flattening multi-file projects into single files and removing code comments, arguing that these steps might oversimplify the task and fail to reflect real-world software engineering scenarios. Although the authors provided ablation studies in the rebuttal to argue that these steps have limited impact, the dissenting reviewer (7ukv) did not respond to confirm if these experiments were sufficient to alleviate the concern about the benchmark's "in-the-wild" representativeness. Furthermore, Reviewer 893y raised a fundamental objection to the "S score" metric, arguing it unfairly favors models with high true positive rates while neglecting false positives, potentially leading to unfair model rankings. While the authors claimed this was based on a misunderstanding of the data, the reviewer did not engage to validate the authors' explanation. Given the severity of a "Strong Reject" (Score 2) based on methodological flaws, the concern is treated as outstanding. Reviewer Nh9b's concerns regarding the lack of failure mode analysis and thread statistics were successfully addressed, leading to a score increase.

**Reviewer Scores:**

Reviewer Nh9b explicitly acknowledged the improvements and raised their score from 6 to 8, indicating full satisfaction with the rebuttal. However, the remaining reviewers did not participate in the discussion period. Consequently, I estimate Reviewer Ht39 would likely maintain their score of 6, as their concerns were partially methodological. Reviewer 7ukv would likely maintain their score of 4, as their critique regarding the lack of agentic baselines and the artificial nature of the benchmark setup was fundamental to the paper's contribution. Most importantly, Reviewer 893y would likely maintain their score of 2; although the authors provided a strong defense regarding the metrics, without the reviewer's confirmation that their understanding was indeed incorrect, the strong objection to the soundness of the evaluation framework stands.

---

### Decision · Program_Chairs · 2026-01-26

Reject